

**All about Nitrite: Exploring Nitrite Sources and Sinks in the Eastern Tropical North**
**Pacific Oxygen Minimum Zone**
John C. Tracey[1, 2], Andrew R. Babbin[3], Elizabeth Wallace[1], Xin Sun[1, 4], Katherine L. DuRussel[1,
5], Claudia Frey[1, 6], Donald E. Martocello III[3], Tyler Tamasi[3], Sergey Oleynik[1], and Bess B.
Ward[1]
[1] Department of Geosciences, Princeton University, Guyot Hall, Princeton, NJ, USA 08544
[2] Department of Biology and Paleo Environment, Lamont Doherty Earth Observatory, Columbia
University, Palisades, NY, USA 10964
[3] Department of Earth, Atmospheric and Planetary Sciences, Massachusetts Institute of
Technology, Cambridge, MA, USA 02138
[4] Department of Global Ecology, Carnegie Institution for Science, Stanford, CA, USA 94305
[5] Department of Civil and Environmental Engineering, Northwestern University, Evanston, IL,
USA 60208
[6] Department of Environmental Sciences, University of Basel, Bernoullistrasse 30, 4056 Basel,
Switzerland
*Correspondence to:* John C. Tracey (jt16@alumni.princeton.edu)



## Abstract

Oxygen minimum zones (OMZs), due to their large volumes of perennially deoxygenated waters, are critical regions for understanding how the interplay between anaerobic and aerobic nitrogen (N) cycling microbial pathways affects the marine N budget. Here we present a suite of measurements of the most significant OMZ N cycling rates, which all involve nitrite ($NO_2^-$) as a product, reactant, or intermediate, in the Eastern Tropical North Pacific (ETNP) OMZ. These measurements and comparisons to data from previously published OMZ cruises present additional evidence that $NO_3^-$ reduction is the predominant OMZ N flux, followed by $NO_2^-$ oxidation back to $NO_3^-$. The combined rates of both of these N recycling processes were observed to be much greater (up to nearly 200x) than the combined rates of the N loss processes of anammox and denitrification, especially in waters near the anoxic / oxic interface. We also show that $NO_2^-$ oxidation can occur in functionally anoxic incubations, measurements that further strengthen the case for truly anaerobic $NO_2^-$ oxidation. We also evaluate the possibility that $NO_2^-$ dismutation provides the oxidative power for anaerobic $NO_2^-$ oxidation. Although almost all treatments returned little evidence for dismutation (as based on product inhibition, substrate stimulation, and stoichiometric hypotheses), results from one treatment under conditions closest to in situ $NO_2^-$ values may support the occurrence of $NO_2^-$ dismutation. The partitioning of N loss between anammox and denitrification differed widely from stoichiometric predictions of at most 29% anammox; in fact, N loss rates at many depths consisted entirely of anammox. Through investigating the magnitudes of $NO_3^-$ reduction and $NO_2^-$ oxidation, testing for anaerobic $NO_2^-$ oxidation, examining the possibility of $NO_2^-$ dismutation, and further documenting the balance of N loss processes, these new data shed light on many open questions in OMZ N cycling research.



## 1. Introduction

Nitrogen (N) is essential for life because of its prominent role in DNA, RNA, and protein chemistry. As a result, N limits biological productivity in many marine environments. The dissimilatory biological N loss and recycling pathways are traditionally understood to be strictly separated by $O_2$ tolerance. The N loss processes of denitrification, the stepwise reduction of $NO_3^-$ to $N_2$, and anaerobic ammonium oxidation (anammox), the oxidation of $NH_4^+$ with $NO_2^-$ to make $N_2$, require low $O_2$ while the N recycling pathways of $NH_4^+$ oxidation to $NO_2^-$ and $NO_2^-$ oxidation to $NO_3^-$ are viewed as obligately aerobic. Importantly, $NO_2^-$ is a product, reactant, or intermediate in all these pathways. Therefore, developing an understanding of $NO_2^-$ sources and sinks is essential for a complete understanding of marine N biogeochemistry.

Oxygen minimum zones (OMZs) and sediments are the two main marine environments where N loss occurs. There are three major OMZs, the Eastern Tropical North Pacific (ETNP), the Eastern Tropical South Pacific (ETSP), and the Arabian Sea, which occupy 0.1 - 1% of total ocean volume, depending on the $O_2$ threshold used (Codispoti and Richards, 1976; Naqvi, 1987; Bange et al., 2000; Codispoti et al., 2005; Lam and Kuypers, 2011). Importantly, the OMZ water column is not completely deoxygenated from top to bottom; OMZs are characterized by an oxygenated surface, a depth interval of steeply declining $O_2$ around the mixed layer depth, called the oxycline, an oxygen deficient zone (ODZ) spanning several hundred meters where $O_2$ declines below the detection limit of common shipboard CTD $O_2$ sensors, and then a second, gradual, oxycline that transitions to oxygenated deep water. Despite OMZ regions' small size, they are responsible for 20-40% of total marine N loss (Brandes and Devol, 2002; Codispoti, 2007; Gruber, 2004), a magnitude significant for the global marine N budget. In this work, in order to answer several open questions about OMZs and marine N cycling, we conducted a suite



75 of $^{15}$N stable isotope measurements of the most important N cycling microbial pathways in

76 OMZs. We report the N loss rates of anammox and denitrification, as well as the N recycling

77 rates of $NO_3^-$ reduction, $NO_2^-$ oxidation, and $NH_4^+$ oxidation, all of which involve $NO_2^-$.

78   A distinctive feature of OMZs is a secondary nitrite maximum (SNM) (Codispoti et al.,

79 2001; Brandhorst, 1959; Codispoti and Packard, 1980). The highest nitrite concentrations within

80 the SNM can reach 10 µM, much higher than the peak values found in the primary nitrite

81 maximum at the base of the photic zone, which average ~100 nM globally (Lomas and

82 Lipschultz, 2006). Several recent works have shown or argued that the SNM's $NO_2^-$ is supplied

83 via high rates of the first step of denitrification, $NO_3^-$ reduction to $NO_2^-$ (Lam et al., 2009; Lam

84 and Kuypers, 2011; Kalvelage et al., 2013; Babbin et al., 2017, 2020). $NO_3^-$ reduction has been

85 proposed (Anderson et al., 1982) to be one-half of a rapid loop where $NO_3^-$ and $NO_2^-$ are

86 recycled through simultaneously occurring $NO_3^-$ reduction and $NO_2^-$ oxidation. This loop has

87 been supported through experimental measurements of both rates (Babbin et al., 2017, 2020;

88 Kalvelage et al., 2013; Lipschultz et al., 1990). In this view, elevated $NO_3^-$ reduction also

89 generates $NH_4^+$, via organic matter (OM) remineralization, which enhances anammox at the

90 expense of denitrification in oxycline and upper ODZ waters (Babbin et al., 2020). In this study,

91 we conducted tests to further document this rapid loop's existence and role in enhancing

92 anammox.

93   Recent measurements of $NO_2^-$ oxidation have returned significant rates from both the

94 oxycline and the ODZ, findings that challenge the paradigm that $NO_2^-$ oxidation is an obligately

95 aerobic process. Evidence for high, widespread $NO_2^-$ oxidation rates in low $O_2$ waters has

96 accumulated from direct rate measurements via $^{15}$N tracers (Füssel et al., 2011; Lipschultz et al.,

97 1990; Peng et al., 2015, 2016; Ward et al., 1989; Kalvelage et al., 2013; Tsementzi et al., 2016;




Sun et al., 2017, 2021; Babbin et al., 2017, 2020), models (Buchwald et al., 2015), and $^{15}$N
natural abundance measurements (Casciotti et al., 2013). Many explanations have been
proposed including microaerophilic nitrite oxidizing bacteria (NOB) adapted to low but non-zero
$O_2$ conditions (Penn et al., 2016; Bristow et al., 2016; Tsementzi et al., 2016; Bristow et al.,
2017) where the $O_2$ for these NOB is transiently supplied to previously deoxygenated waters by
(1) vertical or horizontal mixing of the ocean surface or nearby oxic water (Casciotti et al., 2013;
Tiano et al., 2014; Bristow et al., 2016; Ulloa et al., 2012) or (2) a cryptic $O_2$ cycle where low-
light adapted phototrophs produce $O_2$ that is consumed by NOB (Garcia-Robledo et al., 2017).

While these explanations could account for oxycline and ODZ top observations, they

cannot account for rigorously $O_2$ contamination controlled observations of $NO_2^-$ oxidation in
waters from the deep, dark, and deoxygenated ODZ core (Babbin et al., 2020; Sun et al., 2021).
These ODZ core results are bolstered by sequencing data that show the presence of NOB
exclusive to the deoxygenated ODZ core (Sun et al., 2019) and ODZ core kinetics experiments
where $O_2$ inhibits $NO_2^-$ oxidation (Sun et al., 2021). Here we build on these results by
performing $^{15}$N tracer experiments across a gradient of $O_2$ concentrations, including functionally
anoxic (< 3 nM) $O_2$ concentrations where $O_2$ cannot play a significant biological or
biogeochemical role (Berg et al., 2022).

Anaerobic $NO_2^-$ oxidation would require an alternative oxidant other than $O_2$. Many

candidates have been proposed for this oxidant including $IO_3^-$ (Babbin et al., 2017), $Mn^{4+}$, $Fe^{3+}$
(Sun et al., 2021), the anammox core metabolism (Sun et al., 2021), the observed reversibility of
the nitrite oxidoreductase enzyme (Wunderlich et al., 2013; Kemeny et al., 2016; Koch et al.,
2015; Buchwald and Wankel, 2022), and $NO_2^-$ dismutation (Babbin et al., 2020; Füssel et al.,
2011; Sun et al., 2021). Due to multiple considerations such as very low $IO_3^-$ in the ODZ core



(Moriyasu et al., 2020), low favorability of $Mn^{4+}$ or $Fe^{3+}$ mediated $NO_2^-$ oxidation at marine pH
values (Luther, 2010), low anammox rates that do not explain the observed stoichiometry of
$NO_2^-$ oxidation to anammox (Kalvelage et al., 2013; Babbin et al., 2020; Sun et al., 2021), and
uncertainty if the enzyme hypothesis can account for structural and phylogenetic differences in
the NXRs of the four NOB genera (Buchwald and Wankel, 2022; Sun et al., 2019), we
conducted experiments to test the remaining most plausible hypothesis: $NO_2^-$ dismutation.
$\quad$ $NO_2^-$ dismutation (Eq. (R4)) is energetically favorable (Strohm et al., 2007; Van de
Leemput et al., 2011) although it has not been detected in nature.  The reaction is proposed to
occur in three steps (Eq. (R1-3)) (Babbin et al., 2020) and possible enzymes for steps 2 and 3
(Eqs. (R2, R3)) have been found in ODZ core metagenomic reads and metagenome assembled
genomes (MAGs) (Padilla et al., 2016; Babbin et al., 2020).  While these sequences were not
classified as NOB, they do indicate that parts of the pathway could occur in OMZs.  If
discovered in OMZs, $NO_2^-$ dismutation would be another N loss pathway, albeit one
indistinguishable from denitrification since the $^{15}N$ atoms in $^{30}N_2$ come from $^{15}NO_2^-$ in both
pathways.  Here we evaluate the hypothesis that $NO_2^-$ dismutation is a significant mechanism for
$NO_2^-$ oxidation under low $O_2$, by searching for product inhibition, the inhibition of both $NO_2^-$
oxidation and $^{30}N_2$ production (i.e. denitrification) in response to addition of $NO_3^-$, substrate
stimulation (increases in both $^{30}N_2$ production and $NO_2^-$ oxidation in response to addition of
$^{15}NO_2^-$), and by comparing the $NO_2^-$ oxidation to the produced $^{30}N_2$ ratio.  A ratio near the 3:1
stoichiometry of dismutation (3 $NO_3^-$: 1 $N_2$, Eq. (R4)) would indicate that dismutation could
explain the $NO_2^-$ oxidation measured in the ODZ core.
$3NO_2^- + 2H^+ \rightarrow NO_3^- + 2NO + H_2O$ $\hspace{4cm}$ (R1)
$2NO \rightarrow N_2 + O_2$ $\hspace{6cm}$ (R2)



$2NO_2^- + O_2 \rightarrow 2NO_3^-$      (R3)
$5NO_2^- + 2H^+ \rightarrow N_2 + 3NO_3^- + H_2O$      (R4)

A final area of OMZ biogeochemistry that we investigate is the relative balance between

anammox and denitrification and these pathways' relationships to the rapid $NO_2^-$ oxidation /
$NO_3^-$ reduction loop. After the discovery of anammox, many OMZ studies (Kalvelage et al.,
2013; Kuypers et al., 2005; Hamersley et al., 2007; Jensen et al., 2011; Thamdrup et al., 2006;
Lam et al., 2009), but not all (Ward et al., 2009; Bulow et al., 2010; Dalsgaard et al., 2012) have
reported that anammox is the dominant N loss flux in OMZs, a surprising difference from the
stoichiometric based prediction that OMZ N loss should be at most 29% anammox (Dalsgaard et
al., 2003). This prediction assumes that all $NH_4^+$ for anammox was derived from
remineralization of OM with a mean marine C:N ratio through complete denitrification of $NO_3^-$
to $N_2$ (Dalsgaard et al., 2003, 2012). Anammox rates exceeding 29% of total N loss would
therefore require an additional source of $NH_4^+$ beyond current observations of denitrification and
the resulting $NH_4^+$ remineralization. The best supported explanations for elevated anammox are
that (1) denitrification is the $NH_4^+$ source, but that complete denitrification peaks episodically in
response to OM quality while anammox occurs at a slow, consistent, low rate (Ward et al., 2008;
Thamdrup et al., 2006; Babbin et al., 2014; Dalsgaard et al., 2012). The snapshots afforded by
isotopic incubations on cruises could therefore easily miss episodes of high complete
denitrification. (2) The rapid loop between $NO_3^-$ and $NO_2^-$ described previously functions as an
"engine" to generate $NH_4^+$ for anammox at the expense of denitrification. The observed
magnitudes of $NO_3^-$ reduction and $NO_2^-$ oxidation and these processes' ability to produce $NH_4^+$
from the remineralization of OM with standard C:N ratios without complete denitrification make
this an additional logical hypothesis.





The second hypothesis is supported by several pieces of evidence such as (1)
measurements that the $O_2$ tolerance of $NO_3^-$ reduction and anammox is higher than that of
denitrification and that therefore these processes are more adapted to the oxycline and ODZ top
(Kalvelage et al., 2011; Jensen et al., 2008; Dalsgaard et al., 2014). Additionally, (2) 'omics
studies have revealed widespread incomplete, modular denitrification in OMZs (Sun and Ward,
2021; Ganesh et al., 2015; Fuchsman et al., 2017), and (3) experimental studies have shown that
as $NO_3^-$ reduction increases near the coast, anammox rates also increase (Kalvelage et al., 2013).
According to this view, partial denitrification of $NO_3^-$ to $NO_2^-$ at a much higher rate than
complete denitrification would produce $NH_4^+$ that would then enhance anammox rates. The
resulting enhanced anammox rates occur at the expense of complete denitrification because high
$NO_3^-$ reduction rates would consume OM before the later steps of denitrification. In addition, the
resulting $NO_2^-$ would also be lost to later stage denitrifiers due to high $NO_2^-$ oxidation rates that
would return the $NO_2^-$ to $NO_3^-$. Our study's considerable number of data points, as well as our
ability to compare results to rate measurements obtained from identical methods on previous
cruises offers a unique chance to further validate these explanations for elevated anammox rates.
OMZs are essential regions for the marine N cycle; however, the biogeochemistry of
OMZs may currently be in flux due to anthropogenic pressures. Models and observations
suggest that OMZ volume will grow in the near future, with uncertain impacts (Stramma et al.,
2008; Keeling et al., 2010; Horak et al., 2016; Busecke et al., 2022). As a result, it is important
to develop a thorough understanding of OMZ N cycling to be able to predict any changes in
marine productivity as deoxygenated regions grow. This study contributes towards this goal
through examining four open research questions in OMZ biogeochemistry:



(1) Is the rapid cycle hypothesis correct, i.e., that $NO_3^-$ reduction and $NO_2^-$ oxidation rates are
much greater than N loss rates, especially in the oxycline and ODZ top?
(2) Does truly anaerobic $NO_2^-$ oxidation occur in OMZ regions?
(3) If yes, is $NO_2^-$ dismutation the mechanism by which it occurs?
(4) Is anammox the dominant N loss flux? If yes, what is the explanation?

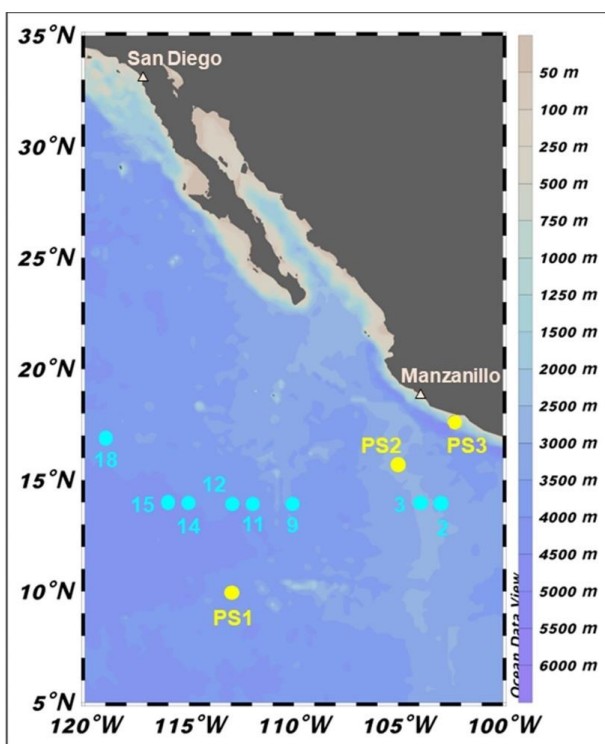


**Figure 1:** Sampling locations during 2018 cruises to the ETNP OMZ. SR1805 stations (spring
2018) are shown in yellow while FK180624 (summer 2018) stations are shown in cyan. Stations
PS1 and 18 are located in more oxic environments on the boundary of the OMZ region. The
remaining FK180624 stations occur along a gradient towards the center of the OMZ region,
represented by stations PS2 and FK180624 stations 2 and 3. These three stations are referred to
as OMZ core stations. Station PS3 (referred to as coastal) represents a final biogeochemical
subregion due to its proximity to the coast.

**2. Methods**
**2.1 $NO_2^-$, $NO_3^-$, and $NH_4^+$ concentration measurements**



Nutrient measurements on all cruises were conducted as follows. Ambient $NO_2^-$
concentrations were measured using the sulfanilimide and NED colorimetric technique with a
spectrophotometer (Strickland and Parsons, 1972). $NO_3^-$ was measured in the $NO_2^-$ oxidation
experiments using the chemilumenscence method (Braman and Hendrix, 1989). Ambient $NH_4^+$
concentrations were measured using the OPA method (Holmes et al., 1999; Taylor et al., 2007;
ASTM International, 2006). In some cases, $NO_2^-$ and $NH_4^+$ were measured on different casts
than those of the rate measurements. In these cases, figures and calculations use interpolated
nutrient values based on the potential density of nutrient sampling and rate measurement depths.
Interpolations were performed with the Matlab pchip function.
**2.2 $NH_4^+$ oxidation and $NO_3^-$ reduction rates**
Incubation experiments were performed on board the R/V *Sally Ride* in March and April
2018 (SR1805). $NH_4^+$ oxidation and $NO_3^-$ reduction rates were measured at three stations: PS1
(open ocean OMZ boundary), PS2 (open ocean, OMZ), and PS3 (coastal OMZ) (Fig. 1). Rates
were measured throughout the water column at ten depths per station (see supplemental Table S1
for depths). Water was directly sampled from the CTD into 60-mL serum vials. After
overflowing three times, bottles were sealed with a rubber stopper and crimped with an
aluminum seal. A 3 mL headspace of He was introduced, samples from below the oxygenated
surface depths were purged for 15 min with He, and then 0.1 mL of tracer solution was added to
all bottles. $^{15}NH_4^+$ and $^{15}NO_3^-$ tracers were added to reach final concentrations of 0.5 μM and 3
μM, respectively. Five bottles were incubated per time course and incubations were ended at 0
(one bottle), 12, and 24 hours (two bottles each) via addition of 0.2 mL of saturated $ZnCl_2$.
Samples were analyzed at the University of Basel using a custom-built gas bench connected by a
Conflow IV interface to a Delta V plus IRMS (Thermo Fisher Scientific). Five mL of the sample





were used to convert $NO_2^-$ to $N_2O$ using the azide method (McIlvin and Altabet, 2005).  A linear
increase of $^{15}N\text{-}NO_2^-$ over time, along with a standard curve to convert from peak area units to
nmol N was used to calculate the $NO_2^-$ production rates according to Eq. (5) and (6) below,
Ammonium oxidation rate $= \dfrac{d\ ^{15}NO_2^-}{dt\ (F_{NH_4^+})}$                                                                     (5)
Nitrate reduction rate $= \dfrac{d\ ^{15}NO_2^-}{dt\ (F_{NO_3^-})}$                                                                        (6)
where:
$\dfrac{d\ ^{15}NO_2^-}{dt}$ is the slope of $^{15}NO_2^-$ produced over time and
$F_{NH_4^+}$ and $F_{NO_3^-}$ are the fraction of the $NO_3^-$ and $NH_4^+$ pools that are labelled with $^{15}N$.
The significance of the rates was evaluated using a Student's t test with a significance level of
0.05.  The reported error bars are the standard error of the regression.  The $NH_4^+$ oxidation rates
reported here were previously published and the experimental method used is more thoroughly
described in this previous publication (Frey et al., 2022).

**2.3 Anammox and denitrification rates depth profiles**

Incubation experiments were performed during SR1805 in March and April 2018 and on

the R/V *Falkor* (FK180624) during June and July 2018.  As above, rates were measured at PS1,
PS2, and PS3 at ten depths per station (see Supplementary Table S1 for sampling depths) during
SR1805.  On FK180624, rates were measured at eight stations that spanned a gradient from the
core of the OMZ region to its edges (see Supplementary Table S2 for sampling depths).  At all
stations and depths water was directly sampled from the CTD into 320 mL borosilicate ground
glass stoppered bottles.  After overflowing three times, bottles were stopped with precision



ground glass caps specifically produced to prevent gas flow. The bottles were transferred to a
glove bag and amended with the following treatments: 3 µM each of $^{15}NO_2^-$ and $^{14}NH_4^+$
(denitrification and anammox) and 3 µM each of $^{15}NH_4^+$ and $^{14}NO_2^-$ (anammox) on SR1805.
Two µM amendments of $^{15}NO_2^-$ and $^{14}NH_4^+$ were used on FK180624. Eight mL of tracer
amended seawater was aliquoted into 12-mL exetainers (Labco). Exetainers were sealed in a
glove bag with butyl septa and plastic screw caps that had been stored under helium for at least
one month, removed and then purged for 5 min at 3 psi with helium gas to remove any $O_2$ that
accumulated during sampling and processing. As a result of this step, it should be noted that all
anammox and denitrification rates sourced from partially or fully oxygenated waters represent
potential rates.

Rates for each sampled depth were calculated using a five-timepoint time course with

three replicates at each point. Incubations were ended by injecting 50 µL saturated $ZnCl_2$ and
vials were stored upside down to prevent the headspace from leaking through the vial cap in
storage and transit. Six months after the cruise, samples were analyzed using a Europa 22-20
IRMS (Sercon). Raw data values were corrected for instrument drift due to run position and
total $N_2$ mass. Drift corrected values and standard curves to convert from peak area units to
nmol $N_2$ were used to calculate rates according to the equations below (Thamdrup et al., 2006;
Thamdrup and Dalsgaard, 2000, 2002) (for more details see supplemental material),
Denitrification (from $^{15}NO_2^-$)
$$\text{Denitrification Rate} = \frac{d\,^{30}N_2}{dt(F_{NO_2^-})^2} \tag{7}$$
Anammox (from $^{15}NO_2^-$)
$$\text{Anammox Rate} = \frac{d\,^{29}N_2}{dt\,F_{NO_2^-}} - 2D\left(1 - F_{NO_2^-}\right) \tag{8}$$





Anammox (from $^{15}NH_4^+$)
$$\text{Anammox Rate} = \frac{d\ ^{29}N_2}{dt\ F_{NH_4^+}}$$    (9)
where:
$\dfrac{d\ ^{30\ or\ 29}N_2}{dt}$ is the slope of the regression of the amount of $^{30\ or\ 29}N_2$ vs. time,
$F_{NO_2^-}$ and $F_{NH_4^+}$ are the fraction of the $NO_2^-$ and $NH_4^+$ pools labelled as $^{15}N$, and
*D* is the denitrification rate calculated according to Eq. (7).
A Student's t test with a significance level of 0.05 was used to evaluate all rates. The reported
error bars are the standard error of the regression. Since the anammox rates measured via both
tracers on the SR1805 cruise were similar in magnitude (Supplementary Table S3), anammox
values reported in Figs. 2, 3, 6, 7, 8, and 9 are based on a combination of these values (see
supplementary material for more information). Previously published (Babbin et al., 2020)
anammox and denitrification rates are sourced from four stations occupied during the R/V
*Thomas G. Thompson*'s March and April 2012 cruise to the ETNP (TN278) and the RVIB
*Nathaniel B. Palmer*'s June and July 2013 ETSP cruise (NBP1305) and were conducted in the
same manner as the SR1805 and FK180624 incubations. Station locations for these cruises were
as follows: TN278 ETNP coastal (20° 00′ N, 106° 00′ W), ETNP offshore (16° 31′ N, 107° 06′
W) and NBP1305 ETSP coastal (20° 40′ S, 70° 41′ W), ETSP offshore (13° 57′ S, 81° 14′ W).

**2.4 SR1805 $NO_2^-$ oxidation depth profiles**

Nitrite oxidation depth profiles were measured in the same exetainers used to measure

anammox and denitrification depth profiles ($^{15}NO_2^-$ treatment only). The rate of $NO_2^-$ oxidation
was determined by converting the $NO_3^-$ produced during the incubations to $N_2O$ using the



denitrifier method (Weigand et al., 2016; Granger, J., & Sigman, 2009) (see supplemental
material for methods details).  The samples were stored at room temperature in the dark until
analysis on a Delta V (Thermo Fisher Scientific) mass spectrometer that measures the isotopic
content of N in $N_2O$ (Weigand et al., 2016).  Samples were corrected for instrument drift due to
run position and total $N_2$ mass (for more details see supplemental materials).  Drift corrected
$\delta^{15}N$ values and a standard curve were then used to calculate the rate as follows,
$$\frac{^{15}N}{^{14}N} = \frac{\left[ \delta^{15}N/1000 + 1 \right] \times 0.003667}{1 - 0.003667} \tag{10}$$

$$NO_2^- \text{ ox. rate} = \frac{d\left[ ^{44}N_2O_{area} \times \,^{15}N/_{14}N \right]}{dt\ F_{NO_2^-}} \tag{11}$$

where Eq. (10) is a rearrangement of the definition of $\delta^{15}N$:
$$\delta^{15}N = \left[ \frac{\frac{^{15}N}{^{14}N}_{sample}}{\frac{^{15}N}{^{14}N}_{air}} - 1 \right] \times 1000 \tag{12}$$

and $^{44}N_2O_{area}$ is the amount of $^{44}N_2O$ measured as sample peak area in V · sec.  0.003667 is the
natural abundance of $^{15}N$ in air.  A Student's t test with a significance level of 0.05 was used to
evaluate all rates.  Reported error bars are the standard error of the regression.  Previously
published (Babbin et al., 2020) $NO_2^-$ oxidation rates are from the previously mentioned TN278
and NBP1305 cruises and were conducted at the same four stations where N loss rates were
measured.  These $NO_2^-$ oxidation rate measurements were conducted according to the same
procedures used for the SR1805 depth profiles.

**2.5 $NO_2^-$ oxidation and $O_2$ manipulation experiments**





Experiments were conducted during cruises SR1805 and FK180624 in spring and
summer 2018. Wide-mouthed Pyrex round media bottles (800 mL total volume, 500 mL
working volume; Corning, USA; product code 1397-500) were used for all incubations. These
bottles were modified to include three stainless steel bulkhead fittings (Swagelok, USA) secured
to the interior of the lid with a Viton rubber gasket and stainless-steel washer between the lid and
the sealing nut. The three ports consisted of two one-eighth inch fluidic ports (inflow and
outflow) and one one-quarter inch sampling port. The fluidic ports were fitted with one-eighth
inch nylon tubing, with the inflow line penetrating to the base of the bottle. The one-quarter inch
sampling port had a butyl rubber septum between the Swagelok stem and nut. This setup
permitted *continuous* gas purging of the bottles while maintaining an otherwise closed system.
For each depth and $O_2$ treatment, three bottles were filled to 500 mL with sample water
and closed. Highly precise digital mass flow controllers (Alicat) were used to establish $O_2$
concentrations in each bottle. Mixing ratios were calculated to create a range of $O_2$
concentrations spanning 1 nM, 10 nM, 100 nM, 1 µM, and 10 µM. The gas mixture modified by
the mass flow controllers was a zero-air gas mixture (Airgas) consisting of 21% $O_2$ and 79% $N_2$
and 1000 ppm $pCO_2$ (the approximate in situ value). Initial gas flow was 1 L min$^{-1}$ for 1 hour to
equilibrate the seawater followed by 100 mL min$^{-1}$ for the remainder of the experiment. Bottles
were daisy-chained together to maintain the same flow rate among them (two bottles on SR1805,
six on FK180624). As in the depth profile experiments, 3 µM $^{15}NO_2^-$ amendments were added
prior to purging. Incubations were conducted in the dark at 12ºC in a cold room (SR1805) or
beverage cooler (FK180624). At the beginning of the experiments, after purging for one hour,
$O_2$ was checked with a LUMOS optode with a detection limit of 0.5 nM (Lehner et al., 2015) and
$CO_2$ was checked by measuring pH using the colorimetric meta-cresol purple method. The



LUMOS optode confirmed that $O_2$ concentrations were within a few nM of the calculated values.
While our use of high precision digital mass flow controllers and this qualitative $O_2$ check
provide confidence that our $O_2$ concentrations are accurate, due to the fact that $O_2$ was not
continuously monitored through the time course, we refer to each $O_2$ concentration as a
"putative" concentration.  Samples (50 mL) were withdrawn every 12 hours for two days with a
four inch hypodermic needle attached to a 60 mL disposable plastic syringe.  Samples were
ejected into acid-cleaned HDPE bottles pre-amended with 200 µL of saturated $ZnCl_2$ solution.
Bottles were screwed closed and wrapped with parafilm.  Samples from each of the three initially
collected bottles were collected to create triplicates at each time point.

**2.6 $NO_2^-$ dismutation experiments**

Nitrite dismutation experiments were performed during SR1805 at Station PS3 (coastal

waters) at two deoxygenated depths: 60 m and 160 m.  Incubations were performed in the same
manner as the above anammox, denitrification, and $NO_2^-$ oxidation experiments where all three
rates were measured in the same exetainers.  Experiments consisted of eight total treatments:
four varying $^{15}NO_2^-$ tracer concentrations (1.125, 5.25, 10.5, and 20.25 µM for 160 m and 0.75,
1.5, 3.75, and 7.5 µM for 60 m) and two $^{14}NO_3^-$ treatments (0 or 20 µM).  As above, both $^{30}N_2$
and $NO_3^-$ production via the denitrifier method (Weigand et al., 2016) were measured.  In order
to test our hypothesis that, if dismutation is occurring, the unexplained $NO_2^-$ oxidation rate (the
difference between the measured $NO_2^-$ oxidation and the $NO_2^-$ oxidation due to anammox) and
the denitrification rate (i.e. the $^{30}N_2$ production rate) should have a 3:1 ratio, a previously
published anammox stoichiometry (Eq. (4) (Kuenen, 2008)) was used to calculate the $NO_2^-$




oxidation due to anammox.  The anammox rates used for this calculation are included in the
supplementary material (Fig. S4).

**360   2.7 Calculation of N loss from $NH_4^+$ oxidation**

The calculation of the maximum possible N loss from $NH_4^+$ oxidation via NO

disproportionation by ammonium oxidizing archaea (AOA) in Supplementary Table S5 was
carried out by dividing the measured $NH_4^+$ oxidation rate by two in accordance with the
stoichiometry of $NH_4^+$ oxidation and NO disproportionation proposed in a previous study (Kraft
et al., 2022).  It should be noted that this operation represents the extreme case where all $^{15}NO_2^-$
produced in $NH_4^+$ oxidation is converted to $N_2$.  We acknowledge this as an unrealistic
assumption used to evaluate the extreme limits of the amount of total N loss attributible to $NH_4^+$
oxidation. This operation was carried out for all depths where $NH_4^+$ oxidation, anammox, and
denitrification rates were measured, irrespective of $O_2$ concentration.

**371   2.8 Redundacy analysis (RDA), Principle component analysis (PCA), and statistics**

All RDA, PCA, redundancy, and correlation analyses were performed with the available

packages in R (v4.2.1 "Funny-Looking Kid") (R: A language and environment for statistical
computing).  All data were first normalized around zero before calculating the Pearson's
correlation coefficient.  Gene abundances (nirS and amoA) used for the RDA and correlation
analyses were measured as previously described (Peng et al., 2015; Jayakumar et al., 2009; Tang
et al., 2022).

**379   2.9 Definition of shallow boundary and ODZ core nomenclature**





In the results and discussion sections, results are classified as shallow boundary or ODZ
core waters according to a previously published threshold (Babbin et al., 2020) where shallow
boundary samples have an in situ potential density < 26.4.  This method is based on a global
profile of OMZ waters meant to delineate shallow boundary samples as waters that are oxic or
may be influenced by $O_2$ intrusions (the surface, the oxycline, and the ODZ top) from those that
are not influenced by $O_2$ instrusions (ODZ core).  It should be noted that a few samples labelled
as ODZ core based on the above criteria are from the deep oxycline waters below the ODZ.

| Depth | $\sigma_\theta$ | OMZ features | $O_2$ intrusions? |
|---|---|---|---|
| Shallow boundary waters | < 26.4 | Surface, oxycline, ODZ top | Yes |
| ODZ core | > 26.4 | ODZ core | No |

**Table 1:** Explanation of shallow boundary waters and ODZ core potential density based
nomenclature (Babbin et al., 2020).
**3. Results**
**3.1 2018 depth profiles of all N cycling rates**
N cycling depth profile experiments were conducted on two cruises (SR1805 and
FK180624) during spring and summer 2018.  These two cruises sampled stations along a
gradient from the edge of the OMZ region to near the coast.  Physical and chemical conditions
varied among stations PS1, PS2, and PS3 on the SR1805 cruise (spring 2018) and across all
FK180624 stations (summer 2018) (Fig. 2, Fig. S1).  Broadly speaking, the vertical span of the
ODZ increased and the top of the ODZ shoaled as distance to shore decreased.  Deep SNM were
observed at almost all stations with the only exceptions being the furthest offshore stations,
stations 11 and 18 from the FK180624 cruise (Fig. S1) and station PS1 from SR1805 (Fig. 2A).
Peak $NO_2^-$ values for all SNM were on the lower side of the range of previous ETNP
observations (Horak et al., 2016), between 1.4 – 2.6 µM.



Of the five N cycling processes measured on the SR1805 cruise, $NO_3^-$ reduction rates had
the greatest magnitude at most depths.  This trend was most pronounced within the upper ODZ,
where $NO_3^-$ reduction rates peaked at station PS2, and the oxycline where $NO_3^-$ reduction rates
peaked at stations PS1 and PS3 (Fig. 2).  Rates of $NO_2^-$ oxidation closely tracked $NO_3^-$ reduction
in distribution; in fact, peak $NO_2^-$ oxidation rates co-occurred with peak $NO_3^-$ reduction rates at








**Figure 2:** SR1805 depth profiles of physical parameters and N cycling rates. **(A)** From left to
right, $O_2$ (µM) and $NO_3^-$ (µM) respectively in blue and black, $NH_4^+$ (nM) and $NO_2^-$ (µM)
respectively in purple and black, temperature (°C) and $\sigma_\theta$ (kg m$^{-3}$) respectively in pink and black,
$NO_2^-$ oxidation and $NO_3^-$ reduction rates (nM N d$^{-1}$) respectively in cyan and black, anammox
and denitrification rates (nM $N_2$ d$^{-1}$) respectively in black and green, $NH_4^+$ oxidation rates (nM
N d$^{-1}$), and percent anammox respectively in coral and black for station PS1 (offshore). **(B)** As
above but for station PS2 (OMZ). **(C)** As above but for station PS3 (coastal). Rates that are
significantly different from zero are shown as filled circles, open circles signify rates not
significantly different from zero. Error bars are the standard error of the regression. Grey dotted
lines indicate upper and lower ODZ boundaries at the time of sampling.
all three SR1805 stations, reaching maxima of ~40 ($NO_2^-$ oxidation) and ~300 ($NO_3^-$ reduction)
nM N d$^{-1}$ at PS3. However, the magnitudes of $NO_2^-$ oxidation rates were usually lower than
$NO_3^-$ reduction rates, sometimes by as much as eightfold. The third N recycling process, $NH_4^+$
oxidation, peaked at or above the oxycline, with peaks of 10 nM N d$^{-1}$ or less. $NH_4^+$ oxidation
was consistently measured to be zero or near-zero throughout the rest of the water column.
Across all SR1805 and FK180624 stations, the magnitude of the N loss processes of
anammox and denitrification was almost always less than 10 nM $N_2$ d$^{-1}$, a much lower magnitude
than the N recycling processes of $NO_3^-$ reduction and $NO_2^-$ oxidation. Like $NO_3^-$ reduction and
$NO_2^-$ oxidation, the two N loss rates peaked in the upper ODZ or right at the oxycline in all three
SR1805 stations, although a deep peak (850 m) in anammox was observed at station PS2 (Fig.
2B). The same pattern was observed in the FK180624 stations with enough coverage of the
entire ODZ water column, stations 2, 9 (6 July sampling), and 9 (9 July sampling) (Fig. S1). The
relative balance between the two N loss processes as measured by percent anammox varied
widely across the water column but largely deviated from the expected partitioning of at most
29% anammox (Dalsgaard et al., 2003, 2012). A striking example of this is that 100% anammox
values were observed in both ODZ core and shallow boundary (see Table 1 for definitions)
samples at many of the SR1805 and FK180624 stations (Fig. 2, Fig. S1).



**3.2 Anaerobic NO$_2^-$ oxidation and O$_2$ manipulation experiments**

Significant NO$_2^-$ oxidation rates were detected in depth profiles across a range of suboxic

O$_2$ concentrations (1 – 5 µM) (definition from (Berg et al., 2022)) across all SR1805 stations,
often at the same depths and in the same vials where the obligately anaerobic processes of
anammox and denitrification were occuring (Fig. 2, Fig. 3A-C, Fig. S2).  In order to
contextualize our observations, we compared our results to previously published measurements
from the TN278 and NBP1305 cruises performed with identical procedures (Babbin et al., 2020).
The highest rates were observed in shallow boundary waters across all three cruises (Fig. 3A-C,
Fig. S2).  Since low but significant levels of O$_2$ can still support aerobic NO$_2^-$ oxidation, a series
of O$_2$ manipulation experiments was carried out on both the SR1805 (spring) and FK180624
(summer) 2018 cruises (Fig. 4A-F and Fig. S3).  In these experiments, where the existence of
functionally anoxic conditions was checked using a LUMOS O$_2$ optode with a detection limit of
0.5 nM (Lehner et al., 2015), we observed significant NO$_2^-$ oxidation, as well as NO$_3^-$ reduction
at putative concentrations as low as 1 nM.  Notably, compared to previous experiments, gas
flushing was constant, with a refresh time of 8 min, so as to maintain O$_2$ levels within the
incubation even while organisms were respiring.  Below 3 nM, O$_2$ is so scarce that such waters
are classified as functionally anoxic, i.e. O$_2$ cannot play biological or biogeochemical roles (Berg
et al., 2022).  As a result, these experiments present convincing additional evidence for the
occurrence of NO$_2^-$ oxidation up to ~100 nM N d$^{-1}$ at O$_2$ concentrations too low to support
aerobic metabolisms.


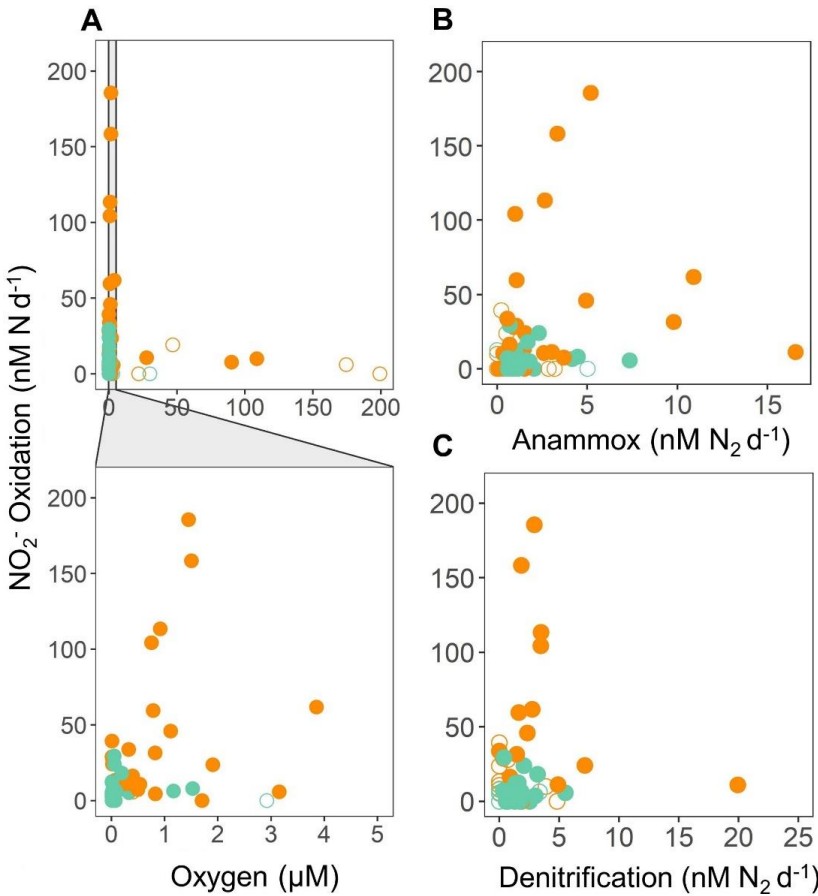

**Figure 3:** NO$_2^-$ oxidation rates (nM N d$^{-1}$) from the 2018 SR1805, TN278 (ETNP 2012), and
NBP1305 (ETSP 2013) cruises vs. **(A)** O$_2$ concentration (μM) from shipboard CTD sensors, **(B)**
anammox rates (nM N$_2$ d$^{-1}$), and **(C)** denitrification rates (nM N$_2$ d$^{-1}$). In A, O$_2$ concentrations
were normalized across cruises. Rates that are significantly different from zero as assessed via a
Student T-test (p value < 0.05) are displayed as filled circles, while insignificant NO$_2^-$ oxidation,
anammox, and dentrification rates are shown as open circles. Rates measured in shallow
boundary waters are colored orange while rates from the ODZ core and below are colored teal.
2012 and 2013 data are republished (Babbin et al., 2020).

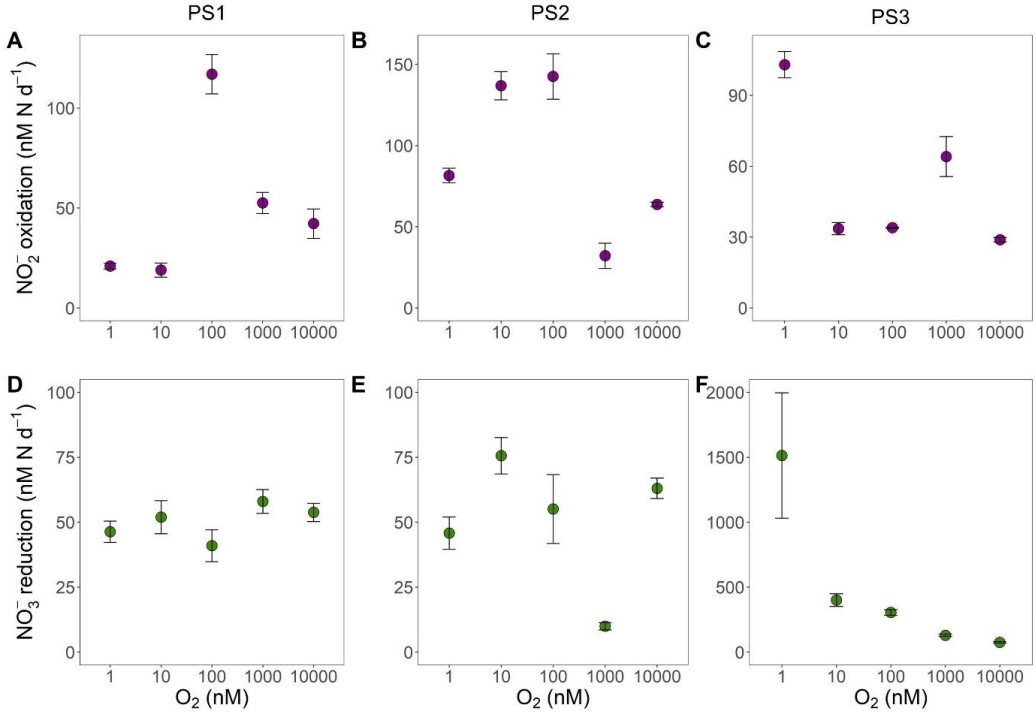

**Figure 4:** Oxygen manipulation experiments that show $NO_2^-$ oxidation (purple) **(A-C)** and $NO_3^-$ reduction (green) **(D-F)** rates (nM N $d^{-1}$) measured across putative $O_2$ concetrations from 1 to 10,000 nM during the SR1805 cruise. Experiments were conducted with waters from the ODZ top: 93 – 110m (PS1) **(A, D)**, 113 – 130m (PS2) **(B, E)**, and 45 – 60m (PS3) **(C, F)**. Error bars are the standard error of the regression. All rates were significantly different from zero.

### 3.3 $NO_2^-$ dismutation

In order to investigate the mechanism for the observed anaerobic $NO_2^-$ oxidation, experiments were conducted to search for evidence of $NO_2^-$ dismutation. If $NO_2^-$ dismutation is the dominant explanation for the observed anaerobic $NO_2^-$ oxidation, we hypothesized that (1) adding $NO_3^-$ should suppress both $^{30}N_2$ and $NO_3^-$ production by LeChatelier's principle, (2) increasing $^{15}NO_2^-$ concentration should increase both denitrification (the $^{30}N_2$ production rate) and $NO_2^-$ oxidation especially when no additional $NO_3^-$ was added, and (3) that the ratio between the "unexplained $NO_2^-$ oxidation," i.e., the difference between the observed $NO_2^-$



oxidation and the $NO_2^-$ oxidation due to anammox, and the observed denitrification ($^{30}N_2$
production) rate should be close to 3:1. In experiments with He-purged water from two
deoxygenated depths (60 and 160 m at station PS3) during the SR1805 cruise we observed that
adding 20 µM $NO_3^-$ suppressed $NO_2^-$ oxidation across nearly all pairs where $NO_2^-$ was identical
and $NO_3^-$ varied between 0 or 20 µM $NO_3^-$ (Fig. 5). However, we did not observe a
simultaneous suppression of $N_2$ production due to the fact that the measured denitrification rate
was low and insignificantly different from zero in most of our 16 treatments (Fig. 5). As a result,
our first hypothesis yielded little evidence of dismutation.

Across all four 60 m 0 µM added $NO_3^-$ treatments (Fig. 5A), adding $NO_2^-$ did increase

$NO_2^-$ oxidation; however, we did not observe an increase in denitrification. Surprisingly, across
the four 60 m 20 µM added $NO_3^-$ treatments, adding $NO_2^-$ decreased $NO_2^-$ oxidation, the reverse
of our hypothesis (Fig. 5). Across all four 160 m 0 µM added $NO_3^-$ treatments, we also observed
an increase in $NO_2^-$ oxidation at higher $NO_2^-$ concentrations but did not observe an increase in
the measured denitrification rate (Fig. 5B). In the four 160 m 20 µM added $NO_3^-$ treatments,
$NO_2^-$ oxidation and denitrification did not increase with $NO_2^-$ concentration (Fig. 5B). Due to
the consistently low and insignificant denitrification rates our test of the $NO_2^-$ addition
hypothesis also yielded little evidence for dismutation.



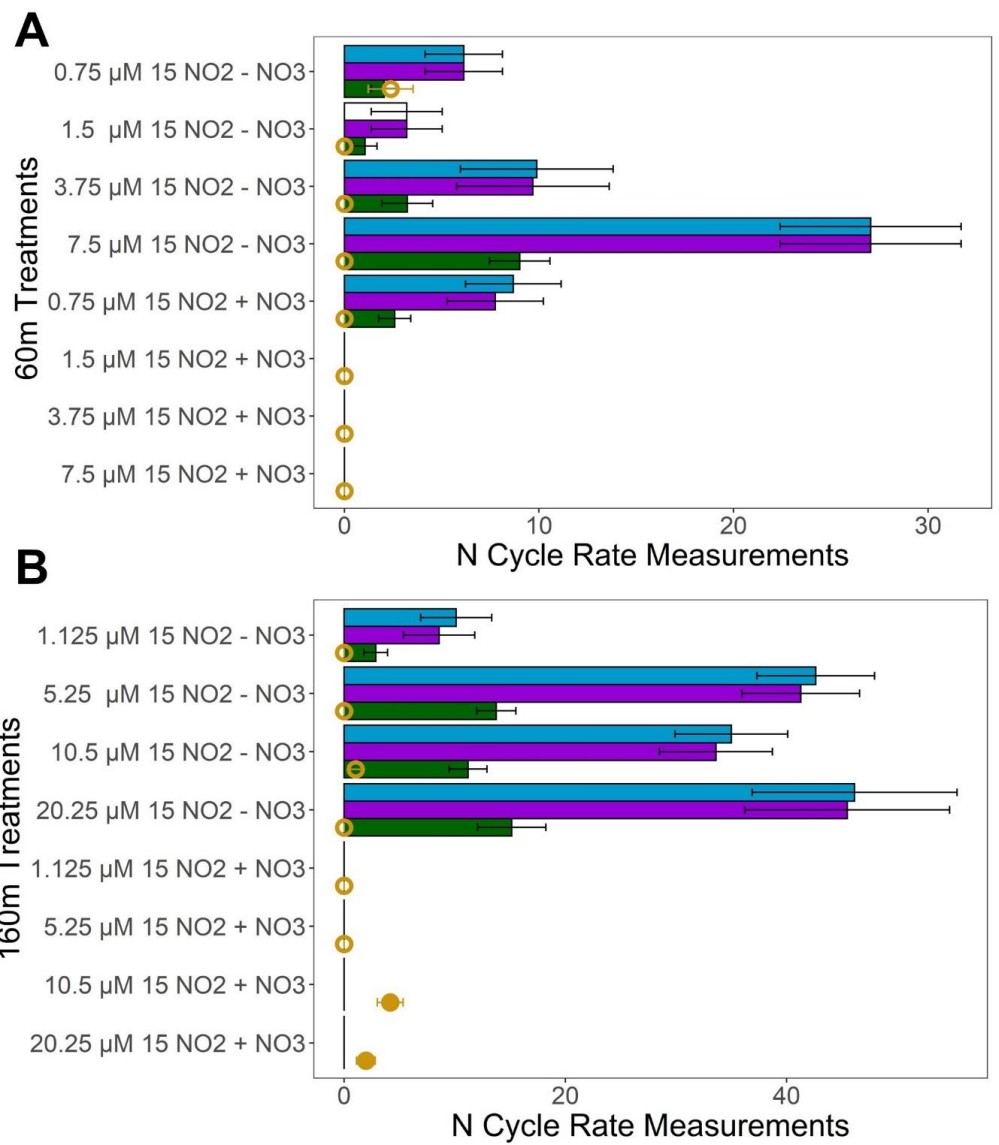

**Figure 5:** $NO_2^-$ dismutation tests conducted in deoxygenated waters from 60m **(A)** and 160m **(B)** at station PS3 during the SR1805 cruise. Measured $NO_2^-$ oxidation rates (nM N $d^{-1}$) are displayed in blue, unexplained $NO_2^-$ oxidation rates, the difference between the measured $NO_2^-$ oxidation and the $NO_2^-$ oxidation due to anammox (nM N $d^{-1}$), are shown in purple. The predicted denitrification (nM $^{30}N_2$ $d^{-1}$) if all the unexplained $NO_2^-$ oxidation was due to $NO_2^-$ dismutation is shown in green. The measured denitrification rate (nM $^{30}N_2$ $d^{-1}$) is shown in yellow where filled circles indicate significant rates and open circles indicate rates that are not significantly different from zero. All bars filled with colors indicate significant rates (i.e. the





white bar for the 60 m 1.5 µM $^{15}NO_2^-$, 0 µM $NO_3^-$ treatment $NO_2^-$ oxidation rate denotes an
insignificant rate). Error bars are the standard error of the regression for $NO_2^-$ oxidation, or are
calculated based on the rules of error propagation from the standard error of the regressions for
the $NO_2^-$ oxidation and anammox rates. (+) $NO_3^-$ treatments received 20 µM $^{14}NO_3^-$ additions
while the (-) $NO_3^-$ treatments received no addition. Anammox rates used to calculate the
unexplained $NO_2^-$ oxidation rate are shown in the supplementary material.

We were also unable to observe evidence for the ratio hypothesis due to the paucity of

significant denitrification ($^{30}N_2$ production) rates (Fig. 5). Since denitrification rates were
consistently low or insignificantly different from zero, the ratio of $NO_2^-$ oxidation to
denitrification deviated from the 3:1 ratio expected if $NO_2^-$ dismutation accounts for most of the
observed $NO_2^-$ oxidation. The only slight exception to this is the 60 m treatment with 0.75 µM
$^{15}NO_2^-$ and 0 µM added $NO_3^-$, the treatment closest to in situ conditions. In this treatment, the
measured denitrification rate, while insignificantly different from zero on the basis of the p value
of the regression, agrees with the predicted denitrification rate based on the 3:1 stoichiometry of
dismutation. While our dismutation experiments as a whole suggest that $NO_2^-$ dismutation is not
a likely explanation for observed anaerobic $NO_2^-$ oxidation, results from the 60 m 0.75 µM
$^{15}NO_2^-$, 0 µM $NO_3^-$ treatment provide slight justification to continue tests of this hypothesis.

**4. Discussion**
**4.1 Rapid $NO_2^-$ / $NO_3^-$ cycle**

Depth profiles of N transformation rates obtained on the SR1805 cruise show that the

rates of $NO_2^-$ oxidation and $NO_3^-$ reduction are far greater than rates of the N loss processes of
anammox and dentrification, especially in shallow boundary (see Table 1 for definition) waters
(Fig. 2, Fig. 6A – B). In fact, when the combined N recycling pathways of $NO_2^-$ oxidation and
$NO_3^-$ reduction are compared to the total N loss, the N recycling pathways are 3.2 – 192.8 times
larger than the total N loss. That the minimum ratio is ~3 strongly emphasizes the



preponderance of $NO_2^-$ oxidation and $NO_3^-$ reduction above N loss processes.  As expected due
to the lower OM concentrations offshore and as previously found in an ETSP N cycling study
(Kalvelage et al., 2013), $NO_2^-$ oxidation and $NO_3^-$ reduction generally increased from the
offshore station (PS1) towards the coast.  We observed $NO_3^-$ reduction rates of a similar
magnitude to previously reported ETSP studies (Kalvelage et al., 2013; Babbin et al., 2017), a
finding that generalizes the predominance of $NO_3^-$ reduction to $NO_2^-$ to the ETNP.  Thus, our
work supports several recent studies (Babbin et al., 2020, 2017; Peters et al., 2016) suggesting
that most nitrogen within OMZ regions is continously recycled between $NO_2^-$ and $NO_3^-$ by rapid
$NO_2^-$ oxidation and $NO_3^-$ reduction, especially in shallow boundary waters.

A previous work (Babbin et al., 2017) predicted that $NO_3^-$ reduction should follow a

Martin curve (Martin et al., 1987) power law distribution across the water column due to its
dependence on the OM flux from shallower waters.  Such a distribution was observed at stations
PS1 and PS3; however, $NO_3^-$ reduction at station PS2 did not follow a classical Martin curve
profile since the $NO_3^-$ production peak is well below the oxycline.  An additional interesting
trend specific to station PS2 is that the deeper peak of $NO_3^-$ reduction coincides with a peak in
complete denitrification to $N_2$ and a steep drop in the percent of N loss due to anammox (Fig. 2).
This connection is also visible in Fig. 6B which shows that $NO_3^-$ reduction increases with total N
loss at station PS2.
These results are consistent with the idea, also supported by many recent studies (Kalvelage et
al., 2013; Lam and Kuypers, 2011; Lam et al., 2009; Babbin et al., 2020, 2017; Füssel et al.,
2011; Lam et al., 2011), that the accumulated $NO_2^-$ in the SNM usually results from an
imbalance between $NO_3^-$ reduction and other N cycling pathways.  We further investigated this
hypothesis by constructing a net $NO_2^-$ budget derived from the five microbial N cycling



metabolisms measured on the SR1805 cruise (Fig. 7). Summing the depth profiles of $NO_2^-$
consumption (anammox, denitrification, and $NO_2^-$ oxidation) and production ($NH_4^+$ oxidation
and $NO_3^-$ reduction) pathways revealed that net depth integrated $NO_2^-$ production across the
sampled OMZ water column depths is on the order of tens of millimoles of $NO_2^-$ per square
meter per day at all three stations (8.19 at PS1, 14.49 at PS2, and 28.97 mmol $NO_2^-$ $m^{-2}$ $d^{-1}$ at
PS3). This excess $NO_2^-$ is driven by $NO_3^-$ reduction, which across all stations is of a much
greater magnitude than all other measured N cycling processes (Fig. 2 and Fig. 7). Additional
support that $NO_3^-$ reduction supplies the accumulated $NO_2^-$ in the SNM can be found by
comparing the net $NO_2^-$ production rates with the measured $NO_2^-$ concentrations along the
SR1805 cruise track from offshore station PS1 to coastal station PS3. As would be expected if
the SNM depended on $NO_2^-$ derived from $NO_3^-$ reduction, the peak net $NO_2^-$ production value
across all depths at each station, the depth integrated $NO_2^-$ production values for each station,
and the magnitude of the SNM peak $NO_2^-$ concentrations all increase together from offshore
station PS1 to coastal station PS3. Importantly, we did not take into account water column
mixing in both vertical and horizontal directions that would carry away produced $NO_2^-$ or $NO_2^-$
assimilation into OM, and we recommend follow up studies that include parameterizations for
these values in OMZ N Cycling modeling.

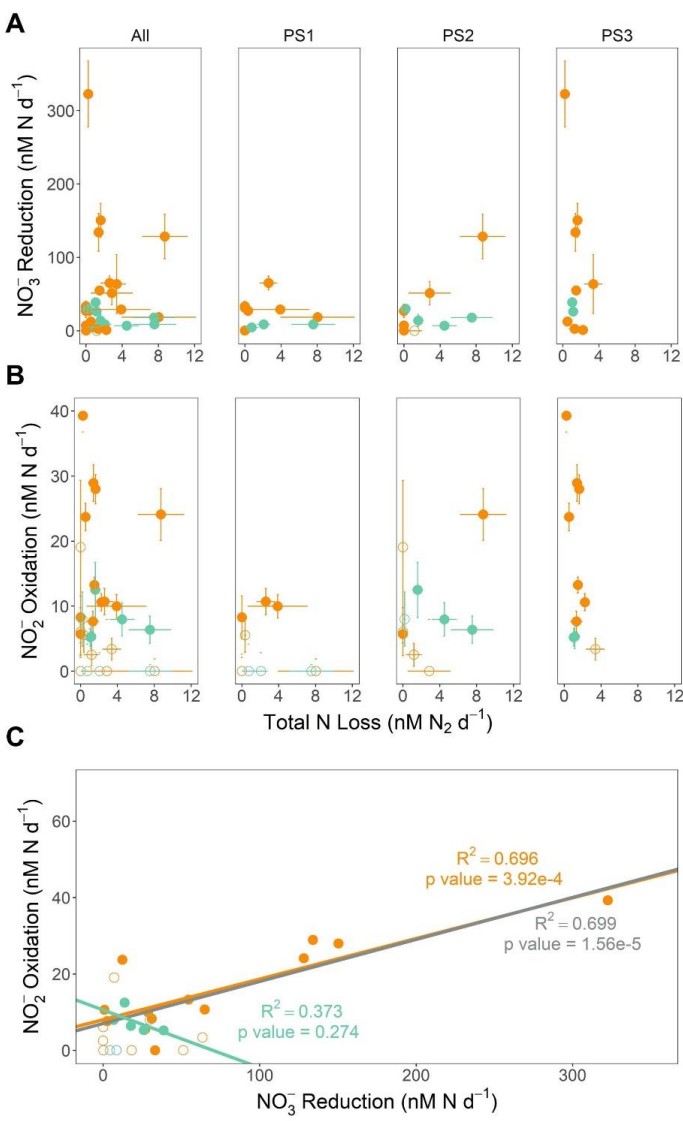


**Figure 6: (A)** NO$_3^-$ reduction (nM N d$^{-1}$) vs. Total N loss (the sum of denitrification and anammox in nM N$_2$ d$^{-1}$) from the SR1805 cruise. **(B)** NO$_2^-$ oxidation (nM N d$^{-1}$) vs. Total N loss from the SR1805 cruise. **(C)** NO$_2^-$ oxidation vs. NO$_3^-$ reduction. Regression lines and statistics are shown for the significant rates from shallow boundary waters only (orange), ODZ core waters only (teal), and all significant data (grey). All points from shallow boundary waters are colored orange while all points from the ODZ core or below are colored teal. Open circles indicate points where the NO$_3^-$ reduction rate (A), NO$_2^-$ oxidation rate (B), or in (C) either NO$_3^-$ reduction or NO$_2^-$ oxidation rate is not significantly different from zero while filled circles indicates rates significantly different from zero.





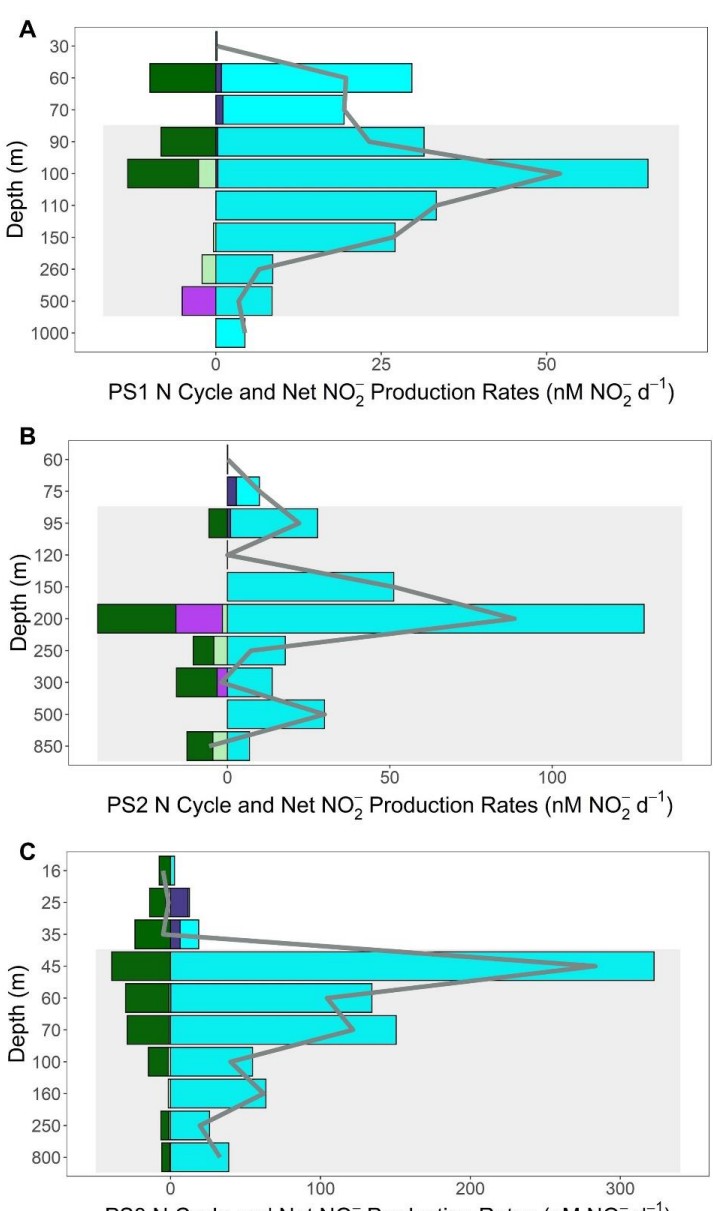

**Figure 7:** $NO_2^-$ budget profiles from the SR1805 cruise. Plots are a combination of the $NO_2^-$ production pathways of $NO_3^-$ reduction (cyan), $NH_4^+$ oxidation (dark purple) and the $NO_2^-$ consumption pathways of anammox (light green), denitrification (bright purple), and $NO_2^-$ oxidation (dark green). Consumption pathways are reported as negative numbers. All rates are reported in nM $NO_2^-$ $d^{-1}$. The net $NO_2^-$ production or consumption rate (nM $NO_2^-$ $d^{-1}$) is represented as a grey line for each depth. Grey boxes indicate the completely deoxygenated ODZ region at each station at the time of sampling. **(A)** PS1, **(B)** PS2, and **(C)** PS3.



**4.2 NO$_2^-$ oxidation – distribution and magnitude in comparison to previous studies**

The high rates of observed NO$_3^-$ reduction provide sufficient NO$_2^-$ to support NO$_2^-$

oxidation both in the oxycline and in the ODZ, as previously proposed (Anderson et al., 1982),
Our observations also further confirm isotopic studies that suggested high NO$_2^-$ oxidation rates
because rapid re-oxidation of NO$_2^-$ back to NO$_3^-$ was necessary to achieve isotopic mass balance
(Buchwald et al., 2015; Casciotti et al., 2013; Granger and Wankel, 2016).  Our results also align
with previous experimental observations of high NO$_2^-$ oxidation rates (Kalvelage et al., 2013;
Babbin et al., 2020; Lipschultz et al., 1990).  Support for a closely connected rapid cycle
between the two processes can be seen in the strong correlation between NO$_2^-$ oxidation and
NO$_3^-$ reduction observed in all SR1805 cruise samples, especially those from shallow boundary
waters (Fig. 6C, Fig. 8).  Similarly to some previous ETSP papers (Babbin et al., 2017, 2020;
Frey et al., 2020) and two ETNP studies (Peng et al., 2015; Sun et al., 2017) we observed that
rates of NO$_2^-$ oxidation, like rates of NO$_3^-$ reduction, peaked in the oxycline or in the ODZ top
(Fig. 2) and then declined throughout the ODZ.  Unlike some stations in these studies (Babbin et
al., 2020, 2017) we did not observe a second peak in NO$_2^-$ oxidation near the deep oxycline.  In
addition to observing a similar distribution, we also observed that NO$_2^-$ oxidation occurs at a
similar magnitude to some stations in previous ETSP studies (Babbin et al., 2020, 2017; Peng et
al., 2016) and ETNP (Peng et al., 2015), although our highest rates (25 – 40 nM N d$^{-1}$) were
much lower than the peaks measured at other stations in most of these reports (Babbin et al.,
2020; Peng et al., 2015, 2016), which reached as high as ~600 nM N d$^{-1}$ (Peng et al., 2015;
Lipschultz et al., 1990).



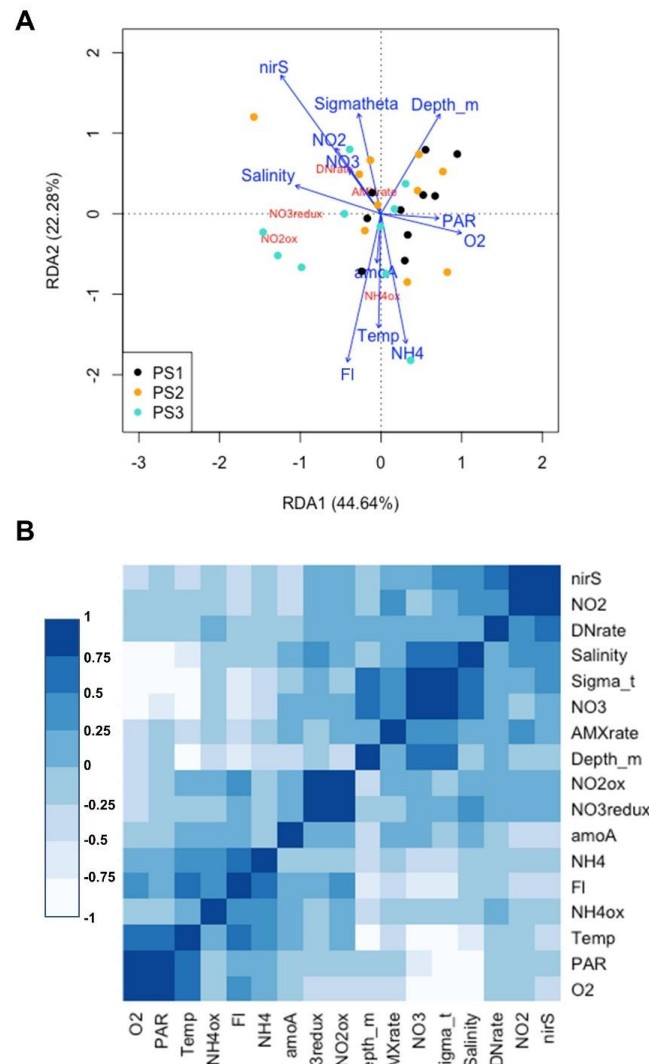


**Figure 8: (A)** Redundancy analysis of all environmental variables and microbial rates measured
on the SR1805 cruise. Points are color-coded by station, black (PS1), yellow (PS2), and cyan
(PS3). Variables names and arrows are color coded so that environmental variables are blue and
rate measurements are red. **(B)** Correlation analysis for all environmental variables and
microbial N cycle rates from the SR1805 cruise. More positive correlations are shaded to
become bluer as significance grows while negative correlations are shaded to become whiter as
significance grows. Abbreviations used are as follows: O2 (oxygen concentration normalized
across different sensors), PAR (photosynthetically active radiation normalized across sensors),
NH4ox ($NH_4^+$ oxidation rate), Fl (chlorophyll fluorescense normalized across different sensors),
NH4 ($NH_4^+$ concentration), amoA (*amoA* abundance), $NO_3^-$redux ($NO_3^-$ reduction rate), NO2ox



(NO$_2^-$ oxidation rate), AMXrate (anammox rate), NO$_3^-$ (NO$_3^-$ concentration), DNrate
(denitrification rate), NO2 (NO$_2^-$ concentration), and nirS (*nirS* abundance).

**4.3 NO$_2^-$ oxidation – can it occur anaerobically?**

NO$_2^-$ oxidation depth profiles (Figs. 2, 3) and O$_2$ manipulation experiments (Fig. 4)

provide further evidence that NO$_2^-$ oxidation can occur under functionally anoxic conditions.
While O$_2$ was not directly measured in the depth profile experiments, several factors argue that
the NO$_2^-$ oxidation observed in these incubations may be O$_2$ independent. As argued previously
(Babbin et al., 2020):
(1) The pre-incubation He purging step in our method removes more than 99% of the N$_2$ present
in exetainers (Babbin et al., 2020). If it is assumed that O$_2$ is removed at identical efficiency, a
reasonable proposition since O$_2$ equilibrates faster than N$_2$ (Wanninkhof, 1992), the introduction
during sample processing of as much as 1 μM O$_2$ would result in a ~10 nM contamination. As a
result, if NO2 oxidation is observed in samples from the deoxygenated ODZ core, contamination
during sampling would be kept very small by our purging step.
(2) Linear timecourses across all timepoints were observed in some of our experiments,
including many from deoxygenated depths at station PS3 (Supplemental Figs. S7-9). If NO$_2^-$
oxidation depended on O$_2$, an initial acceleration (due to O$_2$ contamination that sparked NO$_2^-$
oxidation) or later steep drop (due to the exhaustion of O$_2$ by aerobic NOB) in NO$_2^-$ oxidation
would be expected, not a consistent linear slope.
(3) Metagenomic evidence has revealed distinct NOB communities in oxic surface waters, the
oxycline and ODZ top, and the ODZ core in OMZ regions (Sun et al., 2019). In addition we
observed decreasing NO$_2^-$ oxidation rates with increasing in situ O$_2$ in the SR1805 incubations as
well as the TN278 and NBP1305 incubations (Fig. 3A). These observations are consistent with





the hypothesis that aerobic NOB from oxic depths are ill-equipped to oxidize $NO_2^-$ in
deoxygenated conditions but that the unique MAGs recently identified in draft genomes from the
ODZ top and core (Sun et al., 2019), are adapted to perform anaerobic $NO_2^-$ oxidation.
(4) We observed $NO_2^-$ oxidation at the same depths and often in the same incubation vessels as
the obligately anaerobic processes of anammox and denitrification (Fig. 2, Fig. 3B-C).  Our
observations are consistent with several previous observations that these processes occur at the
same depths (Babbin et al., 2020; Sun et al., 2021).
(5) Through plotting $O_2$ concentrations against the ratio between $NO_3^-$ reduction and $NO_2^-$
oxidation at all SR1805 depths with significant, positive $NO_2^-$ oxidation rates we observed that
the known anaerobic process of $NO_3^-$ reduction and $NO_2^-$ oxidation did not exhibit differential
regulation by $O_2$ as would be expected if $NO_2^-$ oxidation was an obligately aerobic process (Fig.
S5).
Previous studies have shown that $O_2$ additions to purged incubations of ODZ waters
inhibit $NO_2^-$ oxidation (Sun et al., 2017, 2021) and that $NO_2^-$ oxidation can occur in the absence
of $O_2$ consumption (Sun et al., 2021).  However, another kinetics study has reported $O_2$
stimulation of $NO_2^-$ oxidation in OMZ waters (Bristow et al., 2016) and concluded that $NO_2^-$
oxidation is fundamentally an aerobic process.  This apparent contradiction might be explained
by several details in the experimental process of that study:
(1) The study site is at the farthest edge of the ETSP OMZ in a location that is only anoxic in the
austral summer.
(2) The cruise was conducted as austral summer turned to fall (March 20 – 26[th]), a period where
$O_2$ intrusions would be more likely.



(3) $O_2$ data from the study's cruise (Tiano et al., 2014) show that the depths from which $NO_2^-$
oxidation $O_2$ kinetics samples were sourced experienced $O_2$ concentrations of 2 µM (50 m), 10
µM (40 m), and > 60 µM (30m) either during sampling or a few days prior to sampling.
As a result, we argue that the observed stimulation of $NO_2^-$ oxidation by $O_2$ (Bristow et al., 2016)
occurred not because all OMZ NOB are aerobic $NO_2^-$ oxidizers, but instead because the location,
season, and levels of $O_2$ of the sampled station selected for aerobic NOB in the source water for
the purged incubations. Thus, as suggested by (Sun et al., 2017, 2021), different NOB
populations with different historical exposures to $O_2$ and adaptations likely respond differently to
$O_2$ manipulations.

Here we built on the above previous tests of anaerobic $NO_2^-$ oxidation by conducting a

series of incubations across an $O_2$ gradient from 1 nM to 10 µM. Site waters for these
incubations were drawn from the ODZ top at each SR1805 station. We did not observe a clear
inhibitory or stimulatory response of $NO_2^-$ oxidation to $O_2$ within the SR1805 or FK180624
stations, however, this lack of a clear response is in itself a revealing result - a lack of consistent
stimulation by $O_2$ implies at least some anaerobic NOB were present. In addition, we
consistently observed significant $NO_2^-$ oxidation at all putative $O_2$ concentrations, including 1
nM, a functionally anoxic oxygen concentration, i.e., one unable to support aerobic metabolisms
(Berg et al., 2022). Since the initial $O_2$ was supplied by a mass flow controller and subsequently
checked via a very sensitive $O_2$ sensor for all incubations, these results provide additional
evidence that truly anaerobic $NO_2^-$ oxidation can occur.

These $O_2$ manipulation experiments also provided an opportunity to investigate the

response of $NO_3^-$ reduction to $O_2$. The only clear intra-station pattern that emerged from these
experiments was that at station PS3, $NO_3^-$ reduction displayed possible inhibition by $O_2$, as




would be expected. Due to the low number of data points in our data set we did not attempt a
kinetics fitting for this data. Interestingly, the gap observed in depth profile experiments
between the magnitudes of the $NO_3^-$ reduction and $NO_2^-$ oxidation rates was not observed in the
$O_2$ manipulations across many $O_2$ concentrations at stations PS1 and PS2. At station PS3 a large
gap in the magnitudes of these processes as well as the highest overall $NO_3^-$ reduction rates were
observed, as in the depth profile experiments (Fig. 4, 7). A few of the FK180624 data points also
exhibited $NO_3^-$ reduction rates that were elevated far above $NO_2^-$ oxidation (Fig. S3). These
results confirm the importance of $NO_3^-$ reduction for the rapid recycling cycle as well as the
source of $NO_2^-$ for the SNM.

**4.4 $NO_2^-$ dismutation**
In the absence of $O_2$, $NO_2^-$ oxidation would require another oxidant. Many candidate
oxidants have been suggested. For example, iodate ($IO_3^-$), an abundant marine species with
global average marine concentrations of ~0.5 μM (Nozaki, 1997; Lam and Kuypers, 2011), has
been proposed and shown to stimulate $NO_2^-$ oxidation (Babbin et al., 2017). However, since
$IO_3^-$ is usually absent within the ODZ core (Moriyasu et al., 2020), its low concentration makes
$IO_3^-$ mediated anaerobic $NO_2^-$ oxidation unlikely (Babbin et al., 2020). $NO_2^-$ oxidation via $Mn^{4+}$
or $Fe^{3+}$ is thermodynamically feasible, but only at low pH (<6) (Luther, 2010; Luther and Popp,
2002). This pH constraint, combined with the fact that concentrations of these ions are on the
order of a few nM in OMZs (Kondo and Moffett, 2015; Vedamati et al., 2015), makes these
mechanisms unrealistic for the ODZ core. Another proposed mechanism is that the observed
$NO_2^-$ oxidation is due to anammox, which if true should result in an observed $NO_2^-$ oxidation to
anammox ratio of 0.16 – 0.3 (Kuenen, 2008; Strous et al., 1998; Oshiki et al., 2016). Instead, the

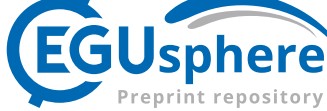

observed ratio is sometimes more than 10x this range and $NO_2^-$ oxidation is rarely observed to be
less than anammox (Kalvelage et al., 2013; Babbin et al., 2020; Sun et al., 2021).
Another alternative hypothesis is based on the reversibility of the nitrite oxidoreductase
(NXR) enzyme.  Since this enzyme has been suggested to both oxidize $NO_2^-$ and reduce $NO_3^-$
(Kemeny et al., 2016; Koch et al., 2015; Wunderlich et al., 2013), $NO_3^-$ reduction by NXR could
over time enrich the $^{15}N$-$NO_3^-$ pool since lighter $^{14}NO_3^-$ would be favored (Casciotti, 2009).
Even in $^{15}NO_2^-$ tracer experiments, in which the $NO_2^-$ pool is highly labeled, this reversibility at
the enzyme site could lead to an apparent transfer of $^{15}N$ from the $NO_2^-$ to the $NO_3^-$ pool if NXR
mediated $NO_3^-$ reduction was occurring.  This hypothesis is supported by observations of $NO_3^-$
reduction under low $O_2$ in cultures from the NOB genera *Nitrobacter* (Freitag et al., 1987; Bock
et al., 1990)*, Nitrospira* (Koch et al., 2015)*,* and in pure cultures of *Nitrococcus mobilis* (Füssel
et al., 2017).  In addition, a recent study presented natural abundance isotopic evidence in pure
*Nitrococcus mobilis* cultures consistent with this mechanism (Buchwald and Wankel, 2022).
However, NXR reversibility has not been demonstrated for the abundant (Füssel et al.,
2011; Mincer et al., 2007) and sometimes predominant (Beman et al., 2013) OMZ NOB genera
*Nitrospina*.  Furthermore, the sole source of the isotopic evidence for the enzyme reversibility
hypothesis, *Nitrococcus mobilis,* has a cytoplasm facing NXR substrate binding domain
(Buchwald and Wankel, 2022), a feature found to have an established evolutionary relationship
to NAR (the known $NO_3^-$ reductase enzyme family) in other *Nitrobacter* studies (Starkenburg et
al., 2008; Kirstein and Bock, 1993).  The NXR substrate binding domains in *Nitrospina* are
oriented towards the periplasm and are not evolutionarily related to enzymes for $NO_3^-$ reduction
(Buchwald and Wankel, 2022; Sun et al., 2019).  Due to these structural and phylogenetic
differences among NOB NXR, it is possible that the *Nitrospina* NXR may be unable to perform



$NO_3^-$ reduction as easily as other NOB genera. For all these reasons, it is not yet clear if the
enzyme reversibility hypothesis can explain all $NO_2^-$ oxidation measured under low $O_2$
conditions and other hypotheses should continue to be explored.

As a result of the above proposals' shortcomings, this paper focused on the remaining,

most plausible hypothesis: $NO_2^-$ dismutation. Our tests for dismutation rested on three
hypotheses: (1) that $NO_3^-$ additions would inhibit both $NO_2^-$ oxidation and $^{30}N_2$ production by
LeChatelier's principle, (2) that increasing $^{15}NO_2^-$ should energetically favor dismutation,
especially in treatments with no additional $NO_3^-$, and (3) that the ratio of non-anammox
mediated $NO_2^-$ oxidation to denitrification ($^{30}N_2$ production) should be close to 3:1 if $NO_2^-$
dismutation explains most of the observed $NO_2^-$ oxidation. We observed repeated inhibition of
$NO_2^-$ oxidation by $NO_3^-$ but no inhibition of $^{30}N_2$ production due to the fact that denitrification
was consistently low and insignificantly different from zero across all treatments. In treatments
with 0 μM added $NO_3^-$, increasing $NO_2^-$ generally increased $NO_2^-$ oxidation, but not
denitrification. In addition, the ratio of anammox corrected $NO_2^-$ oxidation to observed
denitrification deviated from dismutation's 3:1 stoichiometry in almost all treatments. However,
we did observe simultaneous inhibition of $N_2$ and $NO_3^-$ production as well as good agreement
between the anammox corrected $NO_2^-$ oxidation / denitrification ratio to the $NO_2^-$ dismutation
stoichiometry in one treatment - the treatment most similar to in situ conditions (60m, 0.75 μM
$^{15}NO_2^-$, 0 μM $NO_3^-$). As a result, while our results show little evidence for dismutation overall,
we recommend additional experiments at tracer levels similar to 0.75 μM $^{15}NO_2^-$ to further test
for $NO_2^-$ dismutation.

**4.5 Relative balance of anammox and denitrification**





**4.5.1 Are results consistent with past observations of slow, low, and steady anammox**

**elevated above the predicted maximum of 29% of total N loss?**

According to predictions based on the composition of average marine OM (Dalsgaard et al., 2003, 2012) anammox should account for at most 29% of the total N loss flux in OMZ regions. To test this hypothesis under a variety of conditions, regressions of denitrification vs. anammox rates were calculated for all samples from the SR1805, FK180624, TN278, and NBP1305 cruises. In order to compare our new data to a previous study (Babbin et al., 2020), which observed variations in the ratio of anammox and denitrification between samples from the ODZ top or above ($\sigma_\theta < 26.4$, "shallow boundary waters," (Babbin et al., 2020)) and samples from the deoxygenated ODZ core or below ($\sigma_\theta > 26.4$, "ODZ core," (Babbin et al., 2020)), regressions for all data (ODZ core), all data (shallow boundary), 2018 only (ODZ core), 2018 only (shallow boundary), 2012-13 (TN278, and NBP1305) only (ODZ core), and 2012-13 only (shallow boundary) were calculated (Table S4). All regressions deviated from the predicted 29% maximum anammox contour, although the regression from the 2012-13 cruises' ODZ core samples was closest to the 30% anammox contour (Fig. 9A). We observed large differences in the percent anammox contours near 2012-13 and 2018 regressions. ODZ core samples from 2012-13 regressed onto a line between the 40 and 50% anammox contours while ODZ core samples from 2018 regressed onto a line between the 70% and 80% anammox contours. Differences in contouring were smaller for the shallow boundary samples, although the 2018 samples still regressed to a higher contour (just under 80%) than the 2013-13 samples (60%) (Fig. 9A). Our observations that all year and density based regressions fell within contours well above the theoretical prediction (Fig. 9A) and that anammox accounted for as much as 100% of the total N loss at many depths in 2018 samples (Fig. 2, Fig. 9B) is consistent with the many

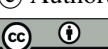



previous studies that observed anammox as the predominant OMZ N loss pathway (Lam et al.,
2009; Thamdrup et al., 2006; Kuypers et al., 2005; Hamersley et al., 2007; Jensen et al., 2011).
Our new 2018 results do not contradict the idea (Dalsgaard et al., 2012) that anammox is
often measured to be the bulk of total N loss but that large, episodic occurences of denitrification
can dwarf the consistent albeit low anammox contribution to total N loss.  Under this view, these
eruptions in denitrification return the *time integrated* balance of anammox and denitrification to
its expected 29 and 71% values.  In this scenario, our cruises' sampling, like many but not all
others, did not coincide with episodic high rates of denitrification.

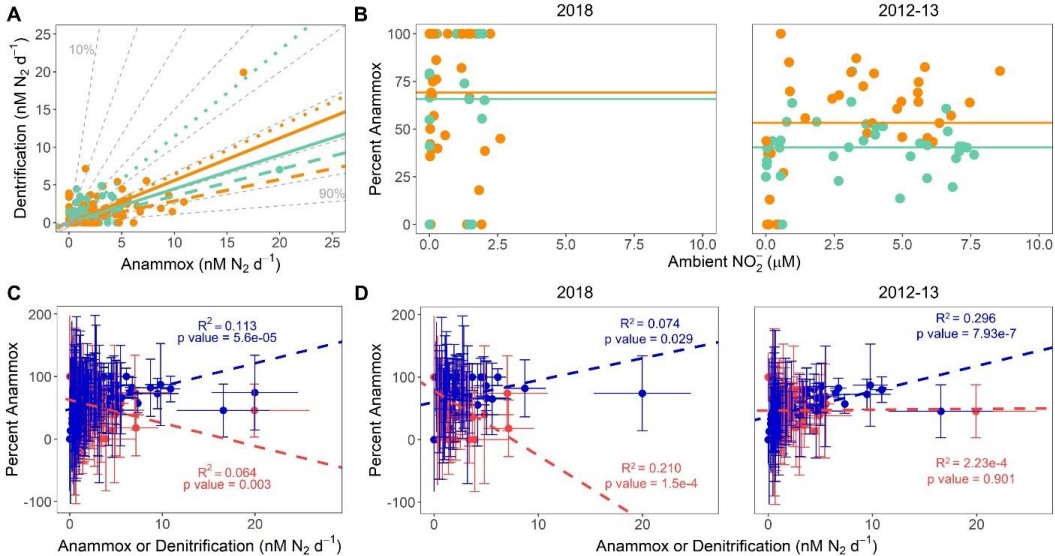


**Figure 9: (A)** All 2012, 2013, and 2018 denitrification and anammox rates (nM $N_2$ $d^{-1}$), color-
coded by $\sigma_\theta$. ODZ core samples and lines are teal ($\sigma_\theta > 26.4$) while shallow boundary samples
and lines are orange ($\sigma_\theta < 26.4$).  Solid, dashed, and dotted lines respectively show regressions
for all data, 2018 only, and 2012-13 data only.  Dashed grey lines depict contours for percent
anammox values. See Supplementary Table S4 for regression statistics. **(B)** Percent anammox
vs. ambient $NO_2^-$ for 2018 samples (left) and republished 2012 and 2013 samples (Babbin et al.,
2020) (right).  Points are colored according to the same scheme as panel A.  Lines show the
average percent anammox values in shallow boundary waters (orange) and the deoxygenated
ODZ core (teal).  **(C)** Percent anammox vs. all anammox (blue) and all denitrification (red) rates
(nM $N_2$ $d^{-1}$).  Regression lines shown for % AMX vs. anammox and denitrification rates follow





the same color scheme as the data points.  Error bars represent the standard error of the
regression.  **(D)** Percent anammox vs. anammox (blue) and denitrification (red) rates (nM $N_2$ $d^{-1}$)
for 2018 only (left) and 2012-13 (right).  Points and regression lines follow the same color
scheme as in panel C.  Data shown in the 2012-13 only panel are republished (Babbin et al.,

2020).



**4.5.2 Do results support a connection between rapid $NO_3^-$ reduction and elevated**
**anammox?**

Our 2018 results question the previously proposed view (Babbin et al., 2020) that rapid

$NO_3^-$ reduction produces $NH_4^+$ that in turn elevates anammox in oxycline and upper ODZ
waters.  While our data (Fig. 2) did find high rates of $NO_3^-$ reduction in shallow boundary
waters, the 2018 N loss data do not show elevated shallow boundary (as compared to ODZ core)
percent anammox values as would be expected if high $NO_3^-$ reduction were fueling elevated
anammox in the oxycline and ODZ top. This difference between our 2018 data and some
previous data (Babbin et al., 2020) in support of a connection between rapid $NO_3^-$ reduction and
elevated anammox in the oxycline and ODZ top can be seen through a comparison of shallow
boundary ($\sigma_\theta$ < 26.4 (Babbin et al., 2020)) and ODZ core ($\sigma_\theta$ >26.4 (Babbin et al., 2020)) percent
anammox values in the 2018 SR1805 and FK180624 cruises against the 2012-13 TN278 and
NBP1305 cruises (Fig. 9B).  2012-13 samples showed a clear partitioning between the ODZ core
and shallow boundary waters in terms of percent anammox values.  In 2012-13, as would be
expected if high oxycline and ODZ top $NO_3^-$ reduction were supplying $NH_4^+$ to anammox,
shallow boundary samples have a higher average percent anammox value than ODZ core
samples (Fig. 9B).  In 2018, this partitioning was not present - the difference between the
average percent anammox values in ODZ core and shallow boundary samples was much smaller
(Fig. 9B).  Interestingly, the total number of samples found to be 100% anammox also sharply
diverged between 2012-13 and 2018.  In the 2012-13 samples, only one shallow boundary



sample was found to be 100% anammox.  In 2018, many samples from both shallow boundary
waters and the ODZ core were 100% anammox (Fig. 9B, Fig. S6).

These observed differences in the partitioning of anammox and denitrification between

shallow boundary waters and the ODZ core across different years and places do not support the
view that $NH_4^+$ from rapid $NO_3^-$ reduction of oxycline and ODZ top OM always elevates
anammox rates.  Instead, they suggest that other factors play an important role in setting the
balance of anammox and denitrification.  Interestingly, $NO_2^-$ concentrations spanned a much
narrower range in the two 2018 SR1805 and FK180624 cruises than the 2012-13 TN278 and
NBP1305 cruises (Fig. 9B), a clue that the biogeochemical environment of the OMZ is subject to
interannual variability.  Observed differences in environmental variables like $NO_2^-$ and percent
anammox partitioning between 2012, 2013, and 2018 suggest that the partitioning of total N loss
must depend on additional yet to be identified environmental or biological  interactions.

**4.5.3 Correlations of percent anammox values to anammox and denitrification rates -**
**comparison to previous literature**

In order to re-examine the result (Babbin et al., 2020) that enhanced fractions of

anammox are correlated to greater anammox rates and not lower dentrification (Fig. 9D right),
we created percent anammox vs. anammox and denitrification regressions with the 2018 SR1805
and FK180624 data.  In 2018, unlike in 2012-13 (Babbin et al., 2020), we observed significant
relationships between percent anammox values and both the anammox and denitrification rates
(Fig. 9D left).  Regressions for the 2012-13 data showed that increases in % anammox values are
correlated only to increases in anammox values, not decreases in denitrification (Babbin et al.,
2020) (Fig. 9D right).  The 2018 regressions, on the other hand, indicate that increases in %



anammox are correlated with both increasing anammox and decreasing denitrification rates. The
influence of this difference in the 2018 samples can be seen in regressions of % anammox
against anammox and denitrification from all three cruises where a similar pattern to the 2018
data is observed (Fig. 9C). As above, this indicates a clear difference in the partitioning of
anammox and denitrification between the 2018 SR1805 and FK180624 ETNP cruises and the
2012-13 TN278 and NBP1305 cruises to the ETNP and ETSP. Despite the significance of the
relationships, the low $R^2$ values indicate that these relationships do not explain most of the
variation in the anammox to denitrification ratio. As above, the causal mechanisms behind this
variability remains to be elucidated.

**4.5.4 Caveats about measurements of anammox and denitrification rates**
One important caveat to some of the above conclusions in section 4.5 is that the detection
limits for anammox and denitrification rates are not identical. It is easier to detect anammox for
a variety of reasons. For example, anammox from a $^{15}NH_4^+$ tracer is more easily detected due to
low background $NH_4^+$ across most of the OMZ. Anammox from the $^{15}NO_2^-$ tracer is more
detectible due to its reliance on incorporation of only a single $^{15}N$ atom into the $^{29}N_2$ product.
Denitrification, on the other hand, is more difficult to detect because of higher background $NO_2^-$
concentrations and because definitive denitrification requires the rarer combination of two
$^{15}NO_2^-$ molecules (Babbin et al., 2017). We suspect that denitrification's higher detection limit
may have played a role in our observations of denitrification rates in the 2012, 2013, and 2018
cruises where, for example, significant denitrification rates were only detected at four of the
thirty depths sampled during SR1805 (Supplementary Table S3). As a result, while the
comparisons made above are helpful to examine differences in N biogeochemistry across years



and stations, the true biogeochemical role of denitrification is likely greater than our tracer
experiments suggest.

### 4.6 Possibility of N loss via AOA and other N cycling processes

A recent paper (Kraft et al., 2022) reported that dense cultures of the ammonium

oxidizing archaea (AOA) *Nitrosopumilus maritimus* can support the $O_2$ dependent process of
$NH_4^+$ oxidation in deoxygenated waters via NO disproportionation to $O_2$ and $N_2$. This
mechanism would be a third N loss process that, if occuring in OMZs, would be measured as
anammox or denitrification. In order to investigate the possible significance of this N loss
pathway in ODZ waters, we calculated the maximum possible N loss from $NH_4^+$ oxidation – the
N loss that would result if all of the $^{15}N\text{-}NO_2^-$ produced in our $NH_4^+$ oxidation experiments was
converted into $N_2$ via the proposed NO disportionation reaction. These maximum $NH_4^+$
oxidation derived N loss rates were a small fraction of the total N loss rates at most depths
(Supplementary Table S5). As a result, even these unrealistically high estimates of $N_2$
production from AOA do not suggest that AOA are significant agents for fixed N loss. The
depths where this was not the case are all either oxic or upper oxycline depths where $NH_4^+$
oxidation rates peak and do not require NO disproportionation to supply $O_2$, or depths where
equally low $NH_4^+$ oxidation, anammox, and denitrification rates would allow a higher percentage
of the total N loss to be due to $NH_4^+$ oxidation. As a result, our calculation argues that N loss
derived from $NH_4^+$ oxidation is not a significant N loss flux in ODZs. Thus, we argue that our
conclusions regarding the relative balance of anammox and denitrification, as well as the
relationship of these two N loss processes to other parts of the N cycle, do not need to be revised
to account for N loss via NO disproportionation in AOA.



We note that an additional N recycling pathway, dissimilatory nitrate/nitrite reduction to
ammonium (DNRA) can occur under low $O_2$ conditions similar to those preferred by anammox
and denitrification.  While some OMZ studies have found rates and *nrfA* abundances comparable
to anammox, denitrification, and $NH_4^+$ oxidation rates and marker gene abundances (Lam et al.,
2009; Jensen et al., 2011), DNRA is best described as an extremely variable process.  Other past
OMZ studies have often found negligible rates (De Brabandere et al., 2014; Kalvelage et al.,
2013; Füssel et al., 2011) and little genetic evidence for DNRA (Kalvelage et al., 2013).  Due to
this variability we chose to focus this study on what are arguably the most consistently relevant
rates for OMZ N biogeochemistry.

**5 Conclusions**
Nitrogen is an essential component of life and as a result, its availability can function as a
cap on biological productivity in many marine ecosystems.  Since all the ocean is linked through
an intricate web of currents that span the globe, the N biogeochemistry of small regions can
affect the biogeochemistry of the rest of the ocean.  Although OMZs account for just 0.1 - 1% of
the ocean's total volume (Lam and Kuypers, 2011; Codispoti and Richards, 1976; Naqvi, 1987;
Bange et al., 2000; Codispoti et al., 2005) they account for 20-40% of all total marine N loss
(Brandes and Devol, 2002; Codispoti, 2007; Gruber, 2004).  As a result, developing an
understanding of N cycling within OMZs is critical for comprehending the total marine N
budget.  Here we presented measurements from the ETNP OMZ of five microbial N cycling
metabolisms, all of which have $NO_2^-$ as a product, reactant, or intermediate.  Understanding the
magnitudes of these rates is key to determining the OMZ inventory of N species as well as an
important piece of understanding the marine N budget.



Our results add to the growing evidence that the N recycling process of $NO_3^-$ reduction is

the largest OMZ N flux followed by the recycling process of $NO_2^-$ oxidation back to $NO_3^-$.
These two processes peaked in the oxycline or ODZ top and were usually much greater than the
two N loss processes of anammox and denitrification, a departure from the established view that
understanding N loss processes alone is the key to understanding OMZ biogeochemistry. We
also add further evidence to the body of literature that supports the occurrence of anaerobic $NO_2^-$
oxidation in OMZ regions, most strikingly through a series of $O_2$ manipulation experiments that
show $NO_2^-$ oxidation at putative $O_2$ concentrations as low as 1 nM, an $O_2$ concentration so low
that the experimental conditions are functionally anoxic. We conducted experiments on waters
from two deoxygenated depths to evaluate if $NO_2^-$ dismutation provides the oxidative power for
observed anaerobic $NO_2^-$ oxidation and found no evidence of $NO_2^-$ dismutation except in one
treatment – the closest to in situ $NO_2^-$ conditions. Further exploration of the dismutation
hypothesis might therefore usefully focus on conditions near in situ $NO_2^-$ concentrations. Across
our experiments, the percent of N loss due to anammox was consistently above the theoretical
prediction of at most 29% anammox. Our observations that $NO_3^-$ reduction and $NO_2^-$ oxidation
greatly surpass N loss, especially in shallow boundary waters, further reinforce the view that
$NO_2^-$ in the SNM is sourced from $NO_3^-$ reduction.

Together, these observations provide additional data that supports several new views of

OMZ biogeochemistry. However, additional work is especially needed to further validate the
occurrence of $NO_2^-$ oxidation under functionally anoxic conditions, explore alternative oxidants
for this process, and comprehend how OMZ biogeochemistry could change with climate change
and other human-caused environmental changes.



**Author contributions**

XS, CF and BBW designed, and CF performed, measured, and calculated the $NO_3^-$ reduction and $NH_4^+$ oxidation rates. BBW and JCT designed, BBW and JCT performed, and JCT measured and calculated the anammox and denitrification depth profile experiments. BBW and XS designed, JCT, BBW, and XS performed, XS and KD measured, and KD, EW, and JCT calculated the $NO_2^-$ oxidation depth profiles. TT and ARB designed, TT performed, DEM and JCT measured, and EW and JCT calculated the anammox and denitrification profiles from the FK180624 cruise. TT and ARB designed, and TT performed, measured, and calculated the $NO_2^-$ oxidation $O_2$ variation experiments. SO provided critical help in running the mass spectrometer to measure all samples except the oxygen variation experiments. BBW performed the correlation and RDA analyses. JCT drafted the paper with inputs from all authors.

**Competing Interests**

The authors declare that they have no conflict of interest.

**Data Availability**

All data discussed in this manuscript will be archived in Zenodo upon publication.

**Acknowledgments**

We would like to acknowledge the crew and scientists of the R/V *Sally Ride* and the R/V *Falkor* for logistical and scientific support during our 2018 cruises. We thank Amal Jayakumar for providing *amoA* and *nirS* gene abundances for the RDA and PCA analyses. We thank Emilio Robledo-Garcia for assistance using the LUMOS sensor for the $NO_2^-$ oxidation $O_2$ gradient



experiments. We thank Matthias Spieler for supporting $NO_3^-$ reduction rate measurements in
Basel. We also acknowledge the Schmidt Ocean Institute which provided R/V *Falkor* ship time
to ARB. We thank the Simons Foundation and National Science Foundation for supporting
ARB, TT, and DEM on Simons Foundation grant 622065 and NSF grants OCE-2138890 and
OCE-2142998 as well as support for BBW, CF, JCT, and XS through NSF grant OCE-1657663.

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
