# Peer review of "All about Nitrite: Exploring Nitrite Sources and Sinks in Oxygen Minimum Zones"

_EGUsphere, 2022_

## Author Response (AR1)

*Note: To structure these comments I included the text of the reviewer's comments in plain text and then inserted our responses in bold beneath each comment or section of the reviewer's comments. All line numbers refer to the clean manuscript copy.

In "All about Nitrite: Exploring Nitrite Sources and Sinks in the Eastern Tropical North Pacific Oxygen Minimum Zone," Tracey et al extensively examine N cycling rates in the ETNP. They performed rates for denitrification, anammox, nitrite oxidation and ammonia oxidation. They also examined nitrite oxidation rates with a constantly purging system that regulated oxygen and used actual sensors to detect the oxygen. To me this addition is a big deal because one of the oxygen treatments was 1 nM. Thus, the authors showed significant nitrite oxidation rates at 1 nM oxygen. This work definitely adds to the literature, but I don't think that they do a good job of selling themself.

I should start this review by admitting that I have not been a believer in anaerobic nitrite oxidation, so this review is written through that lens. However, I think this paper has the first convincing evidence that nitrite oxidation is occurring without oxygen. I think the authors need to shift the focus of their abstract and introduction to highlight this.

Right now, in the introduction, the paper acts like anaerobic nitrite oxidation rates in Babbin et al 2020 are sufficient. If so, why go through so much effort to measure rates a much more complicated manner? The previous rates were not sufficient to convince me because there still could be significant oxygen contamination despite the He bubbling in those types of rates. If I were the authors, I would lay it out in the introduction something like this:

Oxygen contamination is nearly impossible to avoid on deck because 1) Niskin bottles contaminate water at 1-2 uM oxygen before they are ever opened (Garcia-Robledo et al. 2016, 2021). Low oxygen adapted Nitrite oxidizers have a Km for oxygen at something like 4 nM O2 (Bristow et al. 2016). This contrasts to oxygen levels that inhibit denitrification, which are in the hundreds of nanomolar (Dalsgaard et al. 2014).

[Side Note: if the Km for Low oxygen adapted Nitrite oxidizers is 3-4 nM, then 3 nM is not functionally anoxic for nitrite oxidizers. ]

Originally, nitrite oxidation rates were obtained without purging contaminated oxygen (e.g Peng et al 2015, 2016). These rates did match metagenomic abundances of Nitrospina quite well (see Fuchsman et al 2017 Figure 9), so they weren't crazy, but they certainly couldn't be considered anaerobic rates.

In Sun et al 2017, they used a pump system which likely has lower oxygen contamination than Niskin bottles.

Then, rates were obtained after purging with helium in Babbin et al 2020. This was a big step forward. However, oxygen concentrations were not measured in these incubations. Just because oxygen has been reduced to levels where denitrification is not inhibited, does not mean that small amounts of oxygen do not exist. (see differences in important oxygen levels above)

Now here in the current paper, you 1) used a sensor to check oxygen concentrations! And 2) had a crazy set up to purge continuously. This is cool. It needs to be highlighted both in the introduction and the abstract. You have made changes so that more people will believe your results. But they will only read the paper if they know about these changes right away. When I look at the abstract, I see that you are trying to do this. But what you say there could also have been said in Babbin et al 2020—be more specific to make this paper stand out.

**We are very grateful for these suggestions on how to clearly articulate the significance of our oxygen manipulation experiments. We edited the relevant lines in the introduction and abstract (40-42, 110-120) to argue for why our new method (continuous purging and LUMOS sensor deployment) more clearly is a clear advance from previous studies that did not as convincingly address the oxygen contamination known to occur during Niskin sampling.**

**We also addressed the observation from Bristow et al., 2016 that the $k_m$ of nitrite oxidation is in in the low nanomolar range on lines 740 – 751.**

Second issue: The ETNP ODZ (at least offshore) generally has <10 nM ammonium (Widner et al. 2018). You are adding 2 uM ammonium. That is ok, you have to do that to see a signal in the mass spec. But it is more ammonium than these bugs have ever seen before. Additionally, at the top of the ODZ, nitrite is at low concentrations. You are adding 2 uM nitrite. Can you please look at your data and say whether adding ammonium and/or nitrite is stimulating anammox rates? To do this properly, you need to separate rates taken at depth with large in situ nitrite concentration from rates taken at tiny in situ nitrite concentrations and look at the separately.

I will admit that I did this with the rates in Babbin's thesis (Sorry!) and I thought anammox rates were stimulated at the top of the ODZ by nitrite and maybe everywhere by ammonium. If you take this stimulation into account, the anammox rates get significantly smaller. Please make clear the enrichment factor for nitrite and ammonium in the experiments (like ammonium was 200x ambient ammonium). Figure 5 clearly shows that anammox is stimulated by nitrite.

**This concern is a good point. We calculated the factor by which our tracer increased the concentrations of ammonium and nitrite and included these numbers in Table S7. We also noted that due to the enrichment of these nutrients, it is most accurate to conceive of the rates as potential rates (lines 265 – 266, 599). We did not calculate if these enrichments stimulated anammox because no Michaelis-Menten kinetics parameters are available outside of extremely high nutrient and biomass wastewater systems (956 – 959).**

Third issue: It is known that denitrifiers are mostly on particles (Ganesh et al. 2014, 2015; Fuchsman et al. 2017). The way we take samples for rates (small volumes from Niskin bottles) is biased against particles because 1) of the hydrodynamics, CTD rosettes under-sample big particles and 2) the nipple of a Niskin bottle is above the bottom, so big particles sink below the nipple and are not sampled (see (Suter et al. 2017)). Thus, your rates are, for the most part, water column rates. It is a mistake to ignore this.

But the different detection limits of anammox and denitrification is a very good point as well. The 30N2 background in the mass spec can be huge. I think this is the single biggest problem with measuring denitrification. Were different mass specs used between the 2012/2013 and 2018 datasets? Some mass specs are better at detecting 30N2 than others due to the background issue.

**Yes, this concern about particle based denitrification is also a good point. We addressed this caveat in section 4.5.4. We used the same mass spectrometer for the 2012/13 and 2018 measurements.**

Fourth Issue: the abstract. Your potential readers will chose to read or not read this paper based on the abstract, so you want it be compelling. Specifically talk about the upgrades to the nitrite oxidation rate measurements. Lines 43-45 are not very compelling—better to make the section on dismutation short and sweet. Lines 48-51: rewrite in a straight forward way.

**We are grateful for these suggestions on how to make the abstract more compelling.  We reframed the text to emphasize our nitrite oxidation oxygen manipulations,  shortened the section regarding dismutation, and rewrote the last sentence.**

Detailed comments

Line 72: The most recent/accurate citation for this would be 30-50% of N2 production (DeVries et al. 2013)

**We adjusted this citation accordingly.**

Line 105: If you look at (Fuchsman et al. 2019) Figure 2, the Nitrospina depth profile matches the Prochlorococcus depth profile perfectly, supporting the link between primary production and nitrite oxidation in the ODZ.

**We added this citation to the sentence presenting a cryptic oxygen cycle as an explanation for nitrite oxidation.**

Line 106-108: I completely disagree. The methods in Babbin et al 2020 etc, while endeavoring to remove contaminating oxygen, and doing a good job at it, cannot guarantee that oxygen concentrations are below those needed by Nitrite oxidizers. Nitrite oxidizers can use 3 nM O2. That is an order of magnitude lower than levels needed for denitrification to occur. Not faulting the authors. But this technique doesn't guarantee zero oxygen. Best to make clear.

**We rewrote the introduction to address this concern.**

Line 108: There are maybe three Sun et al 2021 papers. I am not sure which one you are referring to here. I think the one not in the references.

**At this line we are referencing Figure 3 from the following paper:**

**https://www.nature.com/articles/s41396-020-00852-3**

**which is the Sun, 2021 ISME Journal paper titled "Microbial niche differentiation explains nitrite oxidation in marine oxygen minimum zones." This study is contained in the references but in reviewing the manuscript again we noticed that it is not clear when the 2021 ISME Journal study vs. the Sun, 2021 ISME Communications study is referenced. We edited the citations to 2021a and 2021b to clarify this.**

Line 110: Please make clear which ODZ you are referencing. Sun et 2019 includes data from the ETSP (~350 m thick ODZ), where the ODZ is hundreds of meters thinner than the ETNP. In the ETNP, where the ODZ was 720 m thick, Nitrospina did go away at depth (Fuchsman et al. 2019). I think it is a good idea to keep clear in your head which studies have data from which ODZ.

**Yes, this is a good point. We were trying to argue that in the Sun, 2019 paper the authors observe that a Nitrospina-like MAG is abundant through the ETSP ODZ to 300m (Figure 4). We acknowledged that this pattern is not seen in other studies or other OMZs by citing the Fuchsman, 2019 paper.**

Line 111: Oxygen doesn't inhibit nitrite oxidation. HIGH concentrations of oxygen inhibit nitrite oxidation. Very big difference. Microaerophiles are a known phenomenon.

**To address this comment, we added the quantitative results from the Sun, 2021 (ISME Journal) study.**

Line 106-114: I think you need to address oxygen intrusions here. It is kind of dismissed at line 105, but there is real data to support oxygen intrusions into the ODZs. For the ETNP see (Margolskee et al. 2019; maybe Monreal et al. 2022).

**We acknowledged these papers' arguments for oxygen intrusions into the ODZ in this section.**

Line 111-114: This sentence is very confusing. You are talking about your experiments, but you cite Berg et al 2022, which is a review paper.

**We cited Berg et al., 2022 because that is the source of our threshold of what constitutes functional anoxia. We removed this part of the sentence and clarified our definition of functional anoxia elsewhere (line 478).**

Line 124: awkward sentence—uncertainty….there is uncertainty??

**We rewrote this sentence along the following lines:**

**"and the enzyme hypothesis' inability to account for structural and phylogenetic differences in the NXRs of the four NOB genera"**

Lines 146-166: Almost all denitrifiers are found on particles (Ganesh et al. 2014, 2015; Fuchsman et al. 2017). The way we take samples for rates (small volumes from Niskin bottles) is

biased against particles because 1) of the hydrodynamics, CTD rossettes undersample big particles and 2) the nipple of a Niskin bottle is above the bottom, so big particles sink below the nipple and are not sampled. Thus, your rates are, for the most part, water column rates. It is a mistake to ignore this.

**See comment above for how we will address this by discussing it in section 4.5.4.**

Lines 146-166: I felt like this section could be reduced.

I think it is particularly important for the authors to understand that the earlier literature on anammox and denitrification rates were all contaminated by oxygen in the stoppers. That is why we now have to keep our stoppers/septa under anaerobic conditions. This was not discovered until Dalsgaard et al 2012—well it was presented at meetings before then. But the 2005-2009 exetainer literature is all contaminated by oxygen. Oxygen inhibits denitrification at lower levels than it inhibits anammox. I think it is best to step carefully here.

**We acknowledge this limitation of early studies on lines 160-161.**

Line 167 and 170: the (1) and (2) in different sentences is awkward.

**We rephrased these sentences to remove the (1) and (2).**

Line 183-187: my favorite current paper related to this is (Ito et al. 2017) which shows that the oxygen content of the Pacific has been decreasing since the 1980's. Also Horak et al is about current changes that actually have happened rather than expected changes. I think that citing current changes (Ito, Horak) is a powerful motivator. Clarify.

**We rephrased this section to emphasize that changes have been observed (Horak and Ito) and that additional future deoxygenation is predicted by models.**

Line 205 etc: Where the measurement performed on the ship or frozen and brought to the lab? Where the samples pre-filtered (if frozen)? This is normal for nitrate concentrations.

**Ammonium and nitrite measurements were performed onboard the ship. Nitrate samples were frozen onboard immediately after collection. Upon return to the Ward laboratory, nitrate samples were thawed and then immediately measured.**

Line 222: What psi of He?

**We added this information in the next draft of the manuscript.**

Line 252: Don't write out the 2 in 2 uM.

**We made this change.**

Annoying comment: Was there N2 gas left in your control samples? This technique assumes that all N2 is purged from the sample before the incubation occurs. Sometimes that is untrue. This is something that you have to data to check and tell the reader.

**In our data processing we accounted for the possibility that a few exetainers contained residual N2 gas by performing an outlier check for each time-course. Outliers above or below three times the mean of the $^{28}N_2$ calculated for each time-course were not used in our calculation of anammox and denitrification rates. The details of this procedure are described on page 1 of the supplemental materials. We changed the text describing the outlier calculation to make it clear that we used this procedure to remove points with some initial $N_2$ from our calculation.**

Line 375-376: Please use the word qPCR when talking about gene abundance measurements to inform the reader about the technique used.

**We made this change.**

Line 385: Calling samples ODZ core that are in the deep oxycline is not acceptable. If they are in the deep oxycline, then call them deep oxycline.

It would make more sense to label depths based on the measured oxygen at those depths.

**The naming scheme for shallow boundary waters and ODZ core waters came from Babbin, 2020 and is based on a global average of the relationship between potential density and oxygen. In line 385 we were trying to state that because the potential density threshold is based on a global average a few depths that in the SR1805 oxygen depth profiles are clearly deep oxycline are labelled as "ODZ core." We edited the sentence to make this clearer. However, we do think that keeping this naming scheme is helpful since it allows comparisons to the Babbin, 2020 paper.**

Line 429: Please tell us what the oxygen concentrations was at 850 m. usually it is still very low at that depth.

**The oxygen concentration was 1.5 µM at 850m – see Table S5.**

Line 430: I am confused by this statement. In Figure S1, station 2 and station 9 (7/6) do not have rates reaching the bottom of the ODZ in the figure.

**We were trying to argue that N loss rates also peaked near the oxycline in these three Falkor stations not that the Falkor stations also had a deep anammox peak. We edited the wording of this sentence to make it clearer.**

All Figures. In all your figures, including supplemental, can you add in minor tick marks or more depth numbers. It is very difficult to determine the depth of a rate from the graph. We should be able to estimates values from a graph.

Figure 3: Please indicate which rate goes with which cruise by shape of the symbol.

**We added minor tick marks to all figures except figures 1, 2, 8, S1, and S7-9. We were unable to add minor tick marks to figures 2 and S1 because the R libraries of ggplot2 and ggh4x do not allow the addition of minor tick marks to a secondary axis. If necessary we can add these tick marks manually in powerpoint for the final revision.**

**We modified the shapes used in figure 3 to show which rate goes with which cruise.**

Figure 4 or line 448: Can you clarify the ambient oxygen concentrations of the water before it was used for these oxygen varying experiments? Maybe in the Figure 4 caption?

**Yes, we added this information as Table S4.**

Line 489: I think it should be noted that ambient nitrate concentrations are often 20 uM in the ETNP ODZ. So when you added nitrate, did you double the nitrate concentrations? It seems to me that the ambient nitrate concentrations are critical for thinking about these experiments. Maybe adding nitrate didn't matter because there was already so much nitrate to begin with. Please tell use the nitrate and oxygen concentrations in the original water for each experiment.

**We added this information on line 517.**

Line 541: (Fuchsman et al. 2019) have productivity measurements in the ETNP offshore, showing it is oligotrophic.

**We added this citation to this section.**

Line 552: If you wanted to, you could mention migrating zooplankton (Bianchi et al. 2014). the biological N2 maximum is largest at similar depths—coincident with the vertical migration depth (Fuchsman et al. 2018). Bianchi is fixated on anammox, but zooplankton also provide organic matter. See (Escribano et al. 2009; Cram et al. 2022). I would guess that this explains your results.

**We added a discussion of migrating zooplankton to this section.**

Line 913: (Fuchsman et al. 2017) found very few reads for genes for DNRA in ETNP metagenomes

**We added this citation to this line.**

Line 924: best citation is DeVries et al 2013

**We added this citation here.**

References

Bianchi, D., A. R. Babbin, and E. D. Galbraith. 2014. Enhancement of anammox by the excretion of diel vertical migrators. Proc. Natl. Acad. Sci. **111**: 15653–15658. doi:10.1073/pnas.1410790111

Bristow, L. A., T. Dalsgaard, L. Tiano, and others. 2016. Ammonium and nitrite oxidation at nanomolar oxygen concentrations in oxygen minimum zone waters. Proc. Natl. Acad. Sci. USA **113**: 10601–6. doi:10.1073/pnas.1600359113

Cram, J. A., C. A. Fuchsman, M. E. Duffy, and others. 2022. Slow particle remineralization, rather than suppressed disaggregation, drives efficient flux transfer through the Eastern Tropical North Pacific Oxygen Deficient Zone. Global Biogeochem. Cycles **36**: e2021GB007080. doi:10.1002/essoar.10507130.1

Dalsgaard, T., F. J. Stewart, B. Thamdrup, L. De Brabandere, P. Revsbech, and O. Ulloa. 2014. Oxygen at Nanomolar Levels Reversibly Suppresses Process Rates and Gene Expression in Anammox and Denitrification in the Oxygen Minimum Zone off Northern Chile. MBio **5**: e01966-14. doi:10.1128/mBio.01966-14.Editor

DeVries, T., C. Deutsch, P. A. Rafter, and F. Primeau. 2013. Marine denitrification rates determined from a global 3-D inverse model. Biogeosciences **10**: 2481–2496. doi:10.5194/bg-10-2481-2013

Escribano, R., P. Hidalgo, and C. Krautz. 2009. Zooplankton associated with the oxygen minimum zone system in the northern upwelling region of Chile during March 2000. Deep. Res. Part II Top. Stud. Oceanogr. **56**: 1083–1092. doi:10.1016/j.dsr2.2008.09.009

Fuchsman, C. A., A. H. Devol, K. L. Casciotti, C. Buchwald, B. X. Chang, and R. E. A. Horak. 2018. An N isotopic mass balance of the Eastern Tropical North Pacific oxygen deficient zone. Deep. Res. Part II Top. Stud. Oceanogr. **156**: 137–142. doi:10.1016/j.dsr2.2017.12.013

Fuchsman, C. A., A. H. Devol, J. K. Saunders, C. McKay, and G. Rocap. 2017. Niche Partitioning of the N cycling microbial community of an offshore Oxygen Deficient Zone. Front. Microbiol. **8**: 2384.

Fuchsman, C. A., H. I. Palevsky, B. Widner, and others. 2019. Cyanobacteria and cyanophage contributions to carbon and nitrogen cycling in an oligotrophic oxygen-deficent zone. ISME J. **13**: 2714–2726. doi:hyyps://doi.org/10.1038/s41396-019-0452-6

Ganesh, S., L. A. Bristow, M. Larsen, N. Sarode, B. Thamdrup, and F. J. Stewart. 2015. Size-fraction partitioning of community gene transcription and nitrogen metabolism in a marine oxygen minimum zone. ISME J. **9**: 2682–2696. doi:10.1038/ismej.2015.44

Ganesh, S., D. J. Parris, E. F. DeLong, and F. J. Stewart. 2014. Metagenomic analysis of size-fractionated picoplankton in a marine oxygen minimum zone. ISME J. **8**: 187–211. doi:10.1038/ismej.2013.144

Garcia-Robledo, E., S. Borisov, I. Klimant, and N. P. Revsbech. 2016. Determination of respiration rates in water with sub-micromolar oxygen concentrations. Front. Mar. Sci. **3**: 1–13. doi:10.3389/fmars.2016.00244

Garcia-Robledo, E., A. Paulmier, S. M. Borisov, and N. P. Revsbech. 2021. Sampling in low oxygen aquatic environments: The deviation from anoxic conditions. Limnol. Oceanogr. Methods **19**: 733–740. doi:10.1002/lom3.10457

Ito, T., S. Minobe, M. C. Long, and C. Deutsch. 2017. Upper ocean $O_2$ trends: 1958–2015. Geophys. Res. Lett. **44**: 4214–4223. doi:10.1002/2017GL073613

Margolskee, A., H. Frenzel, S. Emerson, and C. Deutsch. 2019. Ventilation Pathways for the North Pacific Oxygen Deficient Zone. Global Biogeochem. Cycles **33**: 875–890. doi:10.1029/2018GB006149

Monreal, P. J., C. L. Kelly, N. M. Travis, and K. L. Casciotti. 2022. Identifying the Sources and Drivers of Nitrous Oxide Accumulation in the Eddy-Influenced Eastern Tropical North Pacific Oxygen-Deficient Zone. Global Biogeochem. Cycles **36**: e2022GB007310. doi:10.1029/2022GB007310

Suter, E. A., M. I. Scranton, S. Chow, D. Stinton, L. Medina Faull, and G. T. Taylor. 2017. Niskin bottle sample collection aliases microbial community composition and biogeochemical interpretation. Limnol. Oceanogr. **62**: 606–617. doi:10.1002/lno.10447

Widner, B., C. A. Fuchsman, B. X. Chang, G. Rocap, and M. R. Mulholland. 2018. Utilization of urea and cyanate in waters overlying and within the eastern tropical north Pacific oxygen deficient zone. FEMS Microbiol. Ecol. **94**: fiy138. doi:10.1093/femsec/fiy138

Referee comment #2

The manuscript by Tracey and colleagues presents a detailed study of nitrogen transformations right above and within oxygen deficient waters of the eastern tropical North Pacific. Their work is centered around the assessment of sources and sinks of nitrite as a key component of the nitrogen cycle. Considering pivotal role of nitrogen in the biogeochemistry of low oxygen regions, the topic is timely since the better we understand the spatial-temporal variability of its transformations, the better we can constrain the rather large uncertainties in the global marine nitrogen (and carbon) budgets.

I consider the study novel as it provides provide evidence for nitrite oxidation under anoxic conditions, which is an exciting finding that would explain previous observations within OMZ waters. Likewise, these results set a benchmark upon which future work can build up to refine our understanding of nitrogen cycling under ongoing deoxygenation. It is clear that the experimental approach has limitations in its applicability to in-situ conditions (as acknowledged

by the authors in section 4.5.4.), but showing the importance of a nitrate-nitrite recycling loop is a signifficant finding that challenges our current view on the most important N-cycling processes in marine systems.

The paper is very well written, its structure is clear and the connection between objectives, experimental methods and conclusions is logical and easy to follow. I would only have to minor recommendations with respect to form and content. With regards to the former, I noticed unconsistencies in the spelling of gene names, which by convention are written in italics. With regards to the latter, I did notice that the significance of the study for a wide biogeochemistry community (including colleagues working in observations and modeling) does not come across as clear as the authors might think. I know they mention one or two sentences abuout the general relevance of their findings for the understanding of the nitrogen cycle, but I missed a clear statement on how this exactly can / should be done. In my opinion this comprehensive work should have such a statement to reinforce its added value to the literature.

**Thank you for your comments and suggestions, we are glad to hear that you think the paper is well-written, clear, and significant for the field. We fixed  all gene names to the correct convention and added in a section to the conclusion (lines 1020 – 1028) that notes the significance of the study for the broader biogeochemistry community (modelers and observational scientists).**